# A Robust Functional EM Algorithm for Incomplete Panel Count Data

**Alexander Moreno**
Georgia Institute of Technology
alexander.f.moreno@gatech.edu

**Zhenke Wu**
University of Michigan
zhenkewu@umich.edu

**Jamie Yap**
University of Michigan
jamieyap@umich.edu

**Cho Lam**
University of Utah
cho.lam@hci.utah.edu

**David W. Wetter**
University of Utah
david.wetter@hci.utah.edu

**Inbal Nahum-Shani**
University of Michigan
inbal@umich.edu

**Walter Dempsey**
University of Michigan
wdem@umich.edu

**James M. Rehg**
Georgia Institute of Technology
rehg@gatech.edu

## Abstract

Panel count data describes aggregated counts of recurrent events observed at discrete time points. To understand dynamics of health behaviors and predict future negative events, the field of quantitative behavioral research has evolved to increasingly rely upon panel count data collected via multiple self reports, for example, about frequencies of smoking using in-the-moment surveys on mobile devices. However, missing reports are common and present a major barrier to downstream statistical learning. As a first step, under a missing completely at random assumption (MCAR), we propose a simple yet widely applicable functional EM algorithm to estimate the counting process mean function, which is of central interest to behavioral scientists. The proposed approach wraps several popular panel count inference methods, seamlessly deals with incomplete counts and is robust to misspecification of the Poisson process assumption. Theoretical analysis of the proposed algorithm provides finite-sample guarantees by expanding parametric EM theory [3, 34] to the general non-parametric setting. We illustrate the utility of the proposed algorithm through numerical experiments and an analysis of smoking cessation data. We also discuss useful extensions to address deviations from the MCAR assumption and covariate effects.

## 1 Introduction

A major goal in behavioral medicine is identifying temporal patterns of risk factors preventing an individual from successfully modifying a health-related behavior. In smoking cessation, one would like to know times of day, locations, and other contextual factors such as smoking opportunity [17] that may precipitate lapse to inform interventions to prevent lapse [24]. A basic task is to describe when smoking occurs through modeling the *counting process* of repeated negative events. One goal is to estimate the mean function of the counting process to characterize the dynamics of health-related behaviors at the population level.

The counting process *mean function* can describe population-level temporal patterns of smoking. It converts the discrete patterns of smoking counts from a population of participants (see Fig. 1) into a temporally-continuous summary of smoking behavior (see Fig. 2b). However, the *missingness* inherent in EMA data makes consistent estimation of the mean function difficult. There are several

conditions causing missingness. An EMA may be ignored by the user or opened and then abandoned. Second, the mobile app itself may postpone the triggering of an EMA for any of several reasons (e.g. battery low). While EMAs are triggered randomly, the random process is constrained to have a minimum temporal spacing between EMAs, in order to keep participant burden at an acceptable level. If EMAs are postponed or ignored too many times, it will not be possible to trigger the full set of EMAs for the day, resulting in missing EMAs. Missing data is a common issue in studies that involve EMAs, with [16] noting that over 126 studies, the average missingness rate is 25%.

For mean function estimation, missing EMAs cause problems when they lead to inaccurate counts of the total cigarettes smoked between EMAs. Due to recall bias, longer intervals between EMAs are less reliable.[1] Behavioral scientists have developed heuristic imputation schemes to adjust for missing counts [13], which may not consistently estimate the mean function. We present the first self-contained and systematic treatment of missing data for panel count data, by providing a simple Expectation-Maximization (EM) algorithm [9, 3] to estimate the mean function with finite-sample theoretical guarantees. In doing so, we provide a framework for finite-sample theoretical guarantees for non-parametric EM more generally, potentially allowing finite sample theory for infinite dimensional missing data and latent variable models.

Our primary methodological contribution is a functional EM algorithm to wrap standard non-parametric mean function estimators to handle missing data under a missing completely at random (MCAR) assumption [18]. The E-step uses estimates from a fitting method to impute missing data, and the M-step calls that fitting method to estimate a mean function. This extends several classic non-parametric methods [31, 19] and the baseline-only version of a semi-parametric mean function estimation method [29] to the setting of missing data.

We analyze our EM algorithm using the frameworks described in [3, 34] and obtain finite sample guarantees. This requires care as we extend their work from the parametric to the non-parametric setting. This paper addresses three major theoretical challenges in the context of functional EM by: (i) noting that an infinite dimensional derivative in our setting is analogous to the inner products used in [34, 3], (ii) using a more general technique for showing local uniform strong concavity of our population E-step than previous work, which relied on quadratic dependence on parameters, and (iii) a high probability finite sample bound on the convergence of the M-step. The lack of ground truth for missing data in real-world settings makes evaluation difficult and theoretical guarantees important. Finally, our proposed algorithm can consistently estimate the mean function even when the Poisson process assumption is violated. This is achieved by recovering the population MLE, and noting that under certain integrability conditions the population MLE of the Poisson process log-likelihood is the true mean function of the counting process, even when the counting process is not Poisson.

We analyze two datasets empirically. We first look at a bladder tumor dataset where there are no missing intervals, and simulate missingness under missing completely at random (MCAR), missing at random (MAR), and missing not at random (MNAR) for various initializations and missingness probabilities. We find that recovery under MCAR tends to be very close to if we had complete data, as long as initialization is good. MAR also shows strong performance and MNAR shows relatively strong performance. We further analyze a recent smoking cessation dataset, where the system targets 3-4 prompts per day asking about smoking counts since the last assessment, but sometimes there are fewer responses. We treat self-reports over 24 hours as unreliable or missing data. We find that there is a difference of 21.7% in the final estimates of smoking counts of a 14 day study when using our EM algorithm vs believing their responses over these long intervals.

## 2   Related Work

**Ecological Momentary Assessments** are frequently used to record counts for behavior, including smoking counts [26, 28, 12], alcohol counts [8] and promiscuous behaviors [33]. However, none of these papers use panel count data methods to estimate the mean function.

**Panel count data** analysis comes from nonparametric statistics, but our missing data problem has not been addressed. The closest works to ours are [29, 31, 19]. These papers estimate the mean function

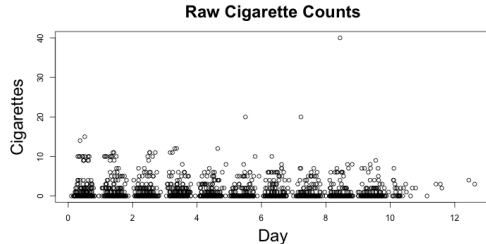

Figure 1: Raw cigarette counts smoked between observations. We want to convert this to a mean function of cumulative cigarettes smoked (Figure 2b), but some counts are missing or unreliable.

from panel count data, but cannot account for missing EMAs as in our motivating application. [29] develop an EM approach under a more limited missingness model assuming that the total counts for each participant are known prior to data analysis. This is unsuitable for analyzing data with missing EMAs. However, two strengths are that they can incorporate baseline covariate information, and they relax the standard assumption of independence between observation times and the counting process.

Neither of [31, 19] handle missing counts, and cannot be applied directly to our setting. We propose an EM algorithm for missing data to wrap any of [29, 31, 19], building on existing methods in the M-step. [31] provides the first strong theoretical guarantees for a mean function estimator in the panel count setting. [19] improve their rate of convergence with spline mean function estimators.

Many other papers in the literature focus on either extending to the semi-parametric setting (which includes baseline covariates), relaxing assumptions in either the non-parametric or semi-parametric setting, or improving estimation under specific distributional assumptions. They do not address missing data. [32] extended [31] to the semi-parametric setting, [15] analyzed panel count with informative observation times and subject-specific frailty, and [14] proposed using a smooth semi-parametric estimator to handle over-dispersion in panel count data. [10] proposed using a squared Gaussian process intensity function.

**Identification in Missing Data Models** a number of recent works [22, 4, 21, 23] analyze when the full law is identified in missing data models, including when recovery is possible under a missing not at random (MNAR) mechanism. However, only [21] discusses estimation, which is our focus. For the parametric non-temporal setting, they give a flexible method for estimation even under MNAR, as long as certain dependency conditions are satisfied by random variables. However, our setting is non-parametric and continuous time, and thus we cannot apply their method.

**EM Theory.** Also relevant is recent work on convergence of the EM algorithm to the true parameter [3, 34], which we extend to the non-parametric setting, proving convergence to the true *mean function*. In [3], they show that under good initialization and certain regularity conditions, population EM converges to the true parameter. With finite sample uniform convergence of the M-step, one can also show a finite-sample bound for EM. [34] show that by assuming the ability to optimize over a ball around the true parameter, one can weaken regularity conditions and replace uniform convergence of the $M$-step with three concentration inequalities, often easier to prove. We base our population guarantees on [34]. We base our finite-sample guarantees on [3], because [34] assumes that empirical and population norms are the same which does not hold in our non-parametric setting.

## 3 Model

### 3.1 Complete Data

Let $N = \{N(u) : u \geq 0\}$ be a univariate counting process. The goal is to estimate the mean function $\Lambda^*(u) = \mathbb{E}[N(u)]$ over a study window $[0, T_{\text{end}}]$. Let $K$ be the random number of observations for a participant. Let $T = (T_1, \cdots, T_K) \in \mathbb{R}_+^K$ be a random vector of observation times, with $T_0 = 0$. Let $\Delta N = (\Delta N_1, \cdots, \Delta N_K)$ be the count increments: $\Delta N_j = N(T_j) - N(T_{j-1})$. Let $\Delta \Lambda(T_j) = \Lambda(T_j) - \Lambda(T_{j-1})$ be mean function increments. For each participant $i = 1, \cdots, n$, we observe $Y = (\Delta N, T, K) \in \mathcal{N} \times \mathcal{T} \times \mathcal{K}$ where $\mathcal{N}, \mathcal{T}, \mathcal{K}$ are the corresponding sample spaces. Let $P_n, P$ denote the empirical and true measures on $\mathcal{N} \times \mathcal{T} \times \mathcal{K}$, respectively. For a measurable function

$f$, let $P_n f = \frac{1}{n} \sum_{i=1}^{n} f(Y_i)$ and $Pf = \int f dP$. Under a non-homogeneous Poisson process, the complete data sample log-likelihood is

$$l_n(Y|\Lambda) = nP_n \left\{ \sum_{j=1}^{K} \Delta N_j \log[\Delta\Lambda(T_j)] - \Lambda(T_K) \right\},$$

and the population log-likelihood is similar but with $P$ instead of $nP_n$. The goal is *consistent mean function estimation even when the Poisson process assumption is violated.*

## 3.2 Missing Data

Unlike previous work, we assume certain observations $\Delta N_j$ are missing. For $\Delta N \in \mathbb{R}^K$, each observation $\Delta N_j$ is missing completely at random with probability (w.p.) $\epsilon \in [0, 1)$. Let $\tau \in \{\circ, 1\}^K$ be the missingness pattern, where $\circ = 0$ but the notation represents missingness. Let $s = 1 - \tau$, thus $s_j = 1$ if $\Delta N_j$ is missing. Let $\Delta N^{(\tau)} = \Delta N \odot \tau$ and $\Delta N^{(s)} = \Delta N \odot s$, where $\odot$ is elementwise product. Then $\Delta N = \Delta N^{(s)} + \Delta N^{(\tau)}$. We observe $\Delta N^{(\tau)}$ but not $\Delta N^{(s)}$. Let $Z \equiv \Delta N^{(s)}$ represent our missing data vector, and let $\mathcal{Z}$ be the space of values for our missing data. Then $Y = (\Delta N^{(\tau)}, T, K)$, and $(Y, Z) \in \mathcal{N} \times \mathcal{T} \times \mathcal{K} \times \mathcal{Z}$ gives us the observed and missing parts of the data, respectively. In this case $P_n$ and $P$ are now the empirical and true measures for $\mathcal{N} \times \mathcal{T} \times \mathcal{K} \times \mathcal{Z}$.

## 4 EM Algorithm

We first describe the general setting for EM. Let $f_\Lambda(y, z)$, $p_\Lambda(y)$, and $k_\Lambda(y|z)$ be joint, marginal, and conditional densities respectively. Let $\Theta$ be a convex set of functions and $\{\Theta_k\}$ a sieve: a nested sequence $\Theta_1 \subset \Theta_2 \subset \cdots$ such that $\cup_{k=1}^{\infty} \Theta_k \subseteq \Theta$ is dense in $\Theta$. We define the following.

**Definition 1.** *(Population Q-functional)* $Q(\Lambda'|\Lambda) \equiv \int_{\mathcal{Y}} \left( \int_{\mathcal{Z}(y)} \log(f_{\Lambda'}(y, z)) k_\Lambda(z|y) dz \right) p_{\Lambda^*}(y) dy.$

**Definition 2.** *(Sample Q-functional)* $Q_n(\Lambda'|\Lambda; \{Y_i\}_{i=1}^{n}) \equiv \frac{1}{n} \sum_{i=1}^{n} \mathbb{E}_\Lambda[\log f_{\Lambda'}(y, z)|Y_i].$

**Definition 3.** $B_r(\Lambda^*) \equiv \{\Lambda : \|\Lambda - \Lambda^*\|_\infty \leq r\}$ *where $\|\cdot\|_\infty$ is the essential supremum.*

**Definition 4.** *(Population M-step)* $M(\Lambda^{(t)}) = \arg\max_{\Lambda' \in \Theta \cap B_r(\Lambda^*)} Q(\Lambda'|\Lambda^{(t)}).$

**Definition 5.** *(Sample M-step)* $M_n(\Lambda^{(t)}) = \arg\max_{\Lambda' \in \Theta_k \cap B_r(\Lambda^*)} Q_n(\Lambda'|\Lambda^{(t)}).$

A $Q$-functional is an $E$-step of the EM algorithm. Population EM repeatedly takes $\Lambda^{(t+1)} = M(\Lambda^{(t)})$, where $\Lambda^{(t)}$ denotes iteration $t$'s estimate. Sample EM repeatedly takes $\Lambda^{(t+1)} = M_n(\Lambda^{(t)})$. Next we describe EM computation. Algorithm 1 describes sample EM; population EM is similar, but uses population $Q$-functionals and $M$-steps.

---

**Algorithm 1** Sample-based EM Algorithm for Panel Count Data with Missing Counts.

---

1: Initialize $\Lambda_0 \in \Theta_k \cap B_r(\Lambda^*)$, let $t \leftarrow 0$
2: **while** not converged **do**
3:     (E-step): Compute $Q_n(\Lambda'|\Lambda^{(t)}; \{Y_i\}_{i=1}^{n})$ using current mean function estimate
4:     (M-step): $\Lambda^{(t+1)} \leftarrow \arg\max_{\Lambda' \in \Theta_k \cap B_r(\Lambda^*)} Q_n(\Lambda'|\Lambda^{(t)}; \{Y_i\}_{i=1}^{n})$ using existing method
5:     $t \leftarrow t + 1$
6: **end while**

---

Algorithm 1 assumes $B_r(\Lambda^*)$ is known following [34]: in practice it is unknown, and we suggest trying multiple initializations and choosing the one that leads to the highest final log-likelihood. In our numerical experiments and data analysis, for illustration we assume $\Theta_k = \{$monotone step functions with at most $k$ steps$\}$. We let $k$ be the total number of intervals across all subjects. Next we describe the E and M-steps.

### 4.1 E-Step

The population and sample $Q$-functionals replace missing counts with mean function estimates. The sample $Q$-functional (the population version is similar but uses $P$ instead of $P_n$) is

$$Q_n(\Lambda'|\Lambda; \{Y_i\}_{i=1}^n) = P_n \left\{ \sum_{j=1}^K \Delta N_j^{\tau_j} \Delta \Lambda^{s_j}(T_j) \log[\Delta \Lambda'(T_j)] - \Lambda'(T_K) \right\}$$

### 4.2 M-Step

The population M-step maximizes $Q(\Lambda'|\Lambda)$ using

$$\Theta = \{\Lambda : [0, T_{\text{end}}] \to [0, \infty) | \Lambda \text{ is nondecreasing}, \Lambda(0) = 0, \Lambda(T_{\text{end}}) \leq U_{\text{all}}\},$$

where $U_{\text{all}}$ is a uniform upper bound for functions in this set. The sample $M$-step uses a convex set $\Theta_k \subset \Theta$ (sieve estimator), where $k \to \infty$ as $n \to \infty$. $\Theta_k$ is potentially monotonic step functions [31, 29] or monotonic splines [19], subject to the constraint of having the upper bound of $U_{\text{all}}$ over the study time. Maximization proceeds via existing methods [31, 19, 29]. Like $B_r(\Lambda^*)$, $U_{\text{all}}$ is unknown.

## 5 Theory

Section 5.1 defines assumptions, 5.2 defines distances and a quasi-inner product, and 5.3 gives two regularity condition lemmas based on [34]. Proposition 1 shows that with good initialization and sufficiently small missingness probability, a population EM step gives a contraction, moving our estimate closer to the true mean function after every iteration. Theorem 1 then shows that population EM converges linearly to the true mean function.

We then show finite-sample theory. Proposition 2 states that with high probability, the sample M-step converges uniformly to the population M-step. Theorem 2 states that with high probability the distance between the current estimate and the true mean function is bounded by two terms: one describes applying population EM, and the other involves the uniform convergence of the $M$-step.

### 5.1 Assumptions

We make the following assumptions, similar assumptions were made in [31, 19, 29]:

1. The counting process $N$ is independent of the number of observations $K$ and observation times $T$, respectively;

2. The observation times are random variables taking values in the bounded set $[\tau_0, T_{\text{end}}]$ where $0 < \tau_0 < T_{\text{end}}$ and $T_{\text{end}} \in (0, \infty)$;

3. The number of observations is bounded, i.e., there exists $k_0 > 0$ such that $P(K \leq k_0) = 1$;

4. The true mean function is uniformly bounded over the study, satisfying $\Lambda^*(u) \leq U \leq U_{\text{all}}$ for some $U \in (0, \infty)$ and all $u \in [\tau_0, T_{\text{end}}]$. Recall $U_{\text{all}}$ is a uniform upper bound on functions in $\Theta$;

5. The first derivative of $\Lambda^*(u)$ has a positive lower bound in $[\tau_0, T_{\text{end}}]$.[2]

6. The observation times are $\alpha$-separated. That is, $P(T_j - T_{j-1} \geq \alpha) = 1$ for some $\alpha > 0$ and all $j = 1, \cdots, K$;

7. $\mathbb{E}[\exp(aN(t))]$ is uniformly bounded for $t \in [0, T_{\text{end}}]$ some constant $a$;

8. The count increments $\Delta N_j$ are missing completely at random (MCAR) w.p. $\epsilon > 0$. That is, $s_{j_i}, j_i = 1, \cdots, K_i, i = 1, \cdots, n$ are iid Bernoulli($\epsilon$) random variables.

In the context of an observational smoking study, these assumptions can be scientifically expressed: 1) EMA delivery times are independent of smoking times 2) EMAs are delivered at random times

throughout the study 3) there is a maximum number of EMAs delivered over the study 5) there is a minimum smoking risk throughout the study 6) there is a minimum time between EMAs 7) a sufficient condition is that the number of cigarettes smoked is bounded over the study 8) the probability that an EMA is missed is $\epsilon > 0$. The only new assumption for panel count is assumption 8.

## 5.2 Measures, Distance Metrics, and Quasi-Inner Product

We next define measures for constructing distance metrics between mean functions. We then define a quasi-inner product that is used to show regularity conditions for population EM theory. Let $B, B_1, B_2$ be the intersection of Borel sets in $\mathbb{R}$ with $[0, T_{\text{end}}]$.

**Definition 6** (Measures for sets containing observation times). $\mu(B) \equiv \mathbb{E}\left[\sum_{j=1}^{K} 1_B(T_j)\right]$ and $\mu_2(B_1 \times B_2) \equiv \mathbb{E}\left[\sum_{j=1}^{K} 1_{B_1}(T_{j-1}) 1_{B_2}(T_j)\right]$.

Then $\mu(B) = \mathbb{E}|\{\text{observations in } B\}|$ and $\mu_2(B_1 \times B_2) = \mathbb{E}|\{\text{one observation in } B_1, \text{ next in } B_2\}|$.

**Definition 7** ($d_2$ metric for mean functions). $\|\Lambda_1 - \Lambda_2\| \equiv [\int_0^{T_{end}} \int_0^{T_{end}} |(\Lambda_1(v) - \Lambda_1(u)) - (\Lambda_2(v) - \Lambda_2(u))|^2 d\mu_2(u, v)]^{1/2}$.

This is the $d_2$ metric of [31]. Under assumption 3, convergence in $\|\cdot\|$ implies convergence in $L^2(\mu)$. We base our theory on convergence in this norm. Now define

$$\langle \nabla Q(\Lambda^{(l)}|\Lambda), \Lambda' \rangle \equiv \lim_{\zeta \downarrow 0} \frac{Q(\Lambda^{(l)} + \zeta \Lambda'|\Lambda) - Q(\Lambda^{(l)}|\Lambda)}{\zeta}$$

$$= P\left\{ \sum_{j=1}^{K} \left( \frac{\Delta N(T_j)^{\tau_j} \Delta \Lambda(T_j)^{s_j}}{\Delta \Lambda_j^l} - 1 \right) (\Delta \Lambda_j') \right\}. \quad (1)$$

We do not claim that (1) is a valid inner product, but it is closely related to inner products in [34, 3], the $Q$-function's directional derivative in the direction of a parameter: (1) is the $Q$-functional's right Gateaux derivative in the direction of a mean function, analogous in function space. The connection to inner products from [3, 34] is key to extending EM theory to the non-paramectric setting, allowing us to prove our Lemmas 1 and 2. We prove Equation (1) in A.1 in the supplementary material.

## 5.3 Population Theory

Before stating our main population EM convergence theorem, we define important constants and state two lemmas for the population $Q$-functional. Lemmas 1 and 2 mirror gradient stability and local uniform strong concavity from [34], respectively, but they studied $Q$-functions with quadratic dependence on parameters. Our objective function does not have quadratic dependence on the mean function, which introduces new challenges. However, we can decompose our $Q$-functional into a sum of two terms: one locally uniformly strongly concave, and one strictly concave. The sum is then locally uniformly strongly concave. Lemma 2 shows this lower bound holds, and is a key step that allows us to apply [34] to our setting. We then prove Proposition 1 which states that population EM steps contract, bringing estimates closer to the true mean function after each iteration.

**Definition 8.** Let $c \equiv \inf \Delta\Lambda^* > 0$ be a uniform lower bound on increments of the true mean function. This exists by assumptions 5 and 6.

**Definition 9.** Let $b \equiv \sup\{\Delta\Lambda : \Lambda \in B_r(\Lambda^*)\} > 0$ be a uniform upper bound on increments of mean functions in $B_r(\Lambda^*)$. Note $b \leq U_{all}$, where $U_{all}$ is a uniform upper bound on functions in $\Theta$.

**Definition 10.** Let $\gamma = \frac{\epsilon}{c}$ and $\nu = \frac{1-\epsilon}{3b}$

**Lemma 1.** (Gradient Stability) Assume assumptions 1, 2, 3, 4, 5, 6, and 8 hold. Then for $\Lambda, \Lambda' \in \Theta$,

$$\langle \nabla Q(\Lambda^*|\Lambda) - \nabla Q(\Lambda^*|\Lambda^*), \Lambda' - \Lambda^* \rangle \leq \gamma \|\Lambda - \Lambda^*\| \|\Lambda' - \Lambda^*\|.$$

*Proof.* See A.2 of the supplementary material. ∎

**Lemma 2.** (Local Uniform Strong Concavity) Assume all assumptions hold. Then if $r \leq \frac{c}{4}$ and $\Lambda', \Lambda \in B_r(\Lambda^*)$,

$$Q(\Lambda^*|\Lambda) - Q(\Lambda'|\Lambda) + \langle \nabla Q(\Lambda^*|\Lambda), \Lambda' - \Lambda^* \rangle \geq \nu \|\Lambda' - \Lambda^*\|^2.$$

*Proof.* See A.3 of the supplementary material. □

One can think of this similarly to a second order Taylor expansion, where we take the functional of $\Lambda'$ expanded at $\Lambda^*$ conditional on $\Lambda$, and use that to derive an inequality.

**Proposition 1.** *(Population EM Contraction) Assume all assumptions hold. If $r \leq \frac{c}{4}$, $\Lambda', \Lambda \in B_r(\Lambda^*)$ and $Q(\Lambda'|\Lambda) \geq Q(\Lambda^*|\Lambda)$, then*

$$\|\Lambda' - \Lambda^*\| \leq \frac{\gamma}{\nu}\|\Lambda - \Lambda^*\|.$$

See section A.5 in the supplementary material for a detailed proof. This is similar to Proposition 3.2 in [34]. In order for this to give a contraction, we need $\gamma < \nu$, which holds if $\epsilon < \frac{c}{3b+c}$. Thus if the uniform lower bound on increments of the true mean function goes up or the uniform upper bound on increments in the ball $B_r(\Lambda^*)$ goes down, we can tolerate a higher probability of missing data. For the smoking setting, these are if the uniform lower bound on mean number of cigarettes smoked between EMAs goes up and uniform upper bound on possible number of cigarettes smoked in the ball around the true mean function goes down.

**Theorem 1.** *(Population EM Convergence to True Mean Function) Suppose the assumptions of the above hold and $0 < \gamma < \nu$. Take the EM sequence $\Lambda^{(t+1)} = \arg\max_{\Lambda' \in \Theta \cap B_r(\Lambda^*)} Q(\Lambda'|\Lambda^{(t)})$ and $\Lambda^0 \in B_r(\Lambda^*) \cap \Theta$. Then*

$$\|\Lambda^{(t)} - \Lambda^*\| \leq \left(\frac{\gamma}{\nu}\right)^t \|\Lambda^{(0)} - \Lambda^*\|.$$

*Proof.* By induction. It holds for $t = 0$. Assume it holds for $t \geq 0$. Then $\Lambda^{(t+1)} \in B_r(\Lambda^*)$ and by assumption $Q(\Lambda^{(t+1)}|\Lambda^{(t)}) \geq Q(\Lambda^*|\Lambda^{(t)})$. Applying population contractivity and the induction assumption,

$$\|\Lambda^{t+1} - \Lambda^*\| \leq \frac{\gamma}{\nu}\|\Lambda^t - \Lambda^*\|$$

$$\leq \left(\frac{\gamma}{\nu}\right)^{t+1} \|\Lambda^0 - \Lambda^*\|$$

□

At each EM step, we move towards the true function by a multiplicative factor of $\left(\frac{\gamma}{\nu}\right)$. This is similar to Theorem 3.1 in [34].

### 5.4 Sample Theory

We next discuss convergence of the sample EM algorithm. We again require the initial estimate $\Lambda^0$ and subsequent estimates $\Lambda^{(t)}$ to be close, i.e., in the set $B_r(\Lambda^*) \cap \Theta_k$. The key additional assumption is that the sample size is large enough to satisfy certain conditions. For all $n$ greater than this minimum sample size, we can guarantee approximate contraction of the sample EM algorithm, where the term that does not contract goes to 0 in large samples.

**Proposition 2.** *Suppose all assumptions hold. Assume $B_r(\Lambda^*)$ has radius $r \leq \frac{c}{4}$. For any $L > 0$, there exists $u_L \overset{L\to\infty}{\to} 0$ such that w.p. $1 - u_L$*

$$\|M_n(\Lambda) - M(\Lambda)\| \leq 2^L n^{-1/3}.$$

*Proof.* See B.1 of the supplementary material. The term $u_L$ is also defined there. □

This is the rate of convergence of an M-estimator, and generalizes the rate of convergence of [1] from maximum likelihood estimates to M-steps. We now state our main result.

**Theorem 2.** *Suppose $0 < r \leq \frac{c}{4}$ and $0 < \gamma < \nu$ and all assumptions hold such that the population contractivity holds. Let $\kappa = \frac{\gamma}{\nu}$. Take the EM sequence $\Lambda^{(t+1)} = \arg\max_{\Lambda' \in B_r(\Lambda^*) \cap \Theta_k} Q_n(\Lambda'|\Lambda^{(t)})$ and $\Lambda^0 \in B_r(\Lambda^*) \cap \Theta_k$. Then if the sample size is large enough that $2^L n^{-1/3} \leq (1 - \kappa)r$, then w.p. at least $1 - u_L$*

$$\|\Lambda^{(t)} - \Lambda^*\| \leq \kappa^t \|\Lambda^0 - \Lambda^*\| + \frac{1}{1-\kappa} 2^L n^{-1/3}.$$

*Proof.* Note that if $\forall \Lambda \in B_r(\Lambda^*) \cap \Theta_k$, $\|M_n(\Lambda) - M(\Lambda)\| \leq 2^L n^{-1/3}$, then we have $\sup_{\Lambda \in B_r(\Lambda^*) \cap \Theta_k} \|M_n(\Lambda) - M(\Lambda)\| \leq 2^L n^{-1/3}$. We claim for any $t > 0$,

$$\|\Lambda^{(t+1)} - \Lambda^*\| \leq \kappa\|\Lambda^{(t)} - \Lambda^*\| + 2^L n^{-1/3}$$

We prove by induction. First, this holds for $t = 1$.

$$\|\Lambda^{(1)} - \Lambda^*\| \leq \|M(\Lambda^{(0)}) - \Lambda^*\| + \|M_n(\Lambda^{(0)}) - M(\Lambda^{(0)})\|$$
$$\leq \kappa\|\Lambda^{(0)} - \Lambda^*\| + 2^L n^{-1/3}$$

Now assume holds true for $t > 0$. Then for $t + 1$,

$$\|\Lambda^{(t+1)} - \Lambda^*\| \leq \|M(\Lambda^{(t)}) - \Lambda^*\| + \|M_n(\Lambda^{(t)}) - M(\Lambda^{(t)})\|$$
$$\leq \kappa\|\Lambda^{(t)} - \Lambda^*\| + 2^L n^{-1/3}$$

Now iterating we have

$$\|\Lambda^{(t)} - \Lambda^*\| \leq \kappa^t\|\Lambda^{(0)} - \Lambda^*\| + \frac{1}{1-\kappa}2^L n^{-1/3}$$

$\square$

The proof follows that of Theorem 5 of [3] closely. This immediately implies that our algorithm can recover the true population MLE in large samples. Further by [31], the population MLE is the true mean function under assumptions 1,4 and 7 even under Poisson process violations.

## 6 Experiments

Here we show numerical and real data results to illustrate the utility of the proposed functional EM algorithm. In the M-step, we can choose any reasonable mean function estimator. In our experiments, we use a general likelihood-based augmented estimating equation (AEE) method [29], which uses monotone step functions when obtaining the mean function estimate with complete data. [30, 7] provide software implementations of AEE and a wide variety of other panel count methods. First, we perform synthetic experiments that demonstrate accurate recovery of the true mean function (see Section C of the supplemtary material). In particular, we also demonstrate good recovery of the true mean function using simulated data based on mixed Poisson processes (which violate the Poisson process assumption) hence confirming theoretical robustness of functional EM to Poisson process assumption. We also show a simulation where simple baselines (complete case analysis, median imputation, and last value carried forward) perform poorly. In the following, we focus on two experiments involving real data.

### 6.1 Real Data with Synthetic Missingness

We analyze blaTum (bladder tumor dataset) [5] with 85 patients and counts of tumors taken at appointment times. We artificially delete intervals completely at random with probability $0.2$. We then initialize $\Lambda^{(0)}$ by replacing the missing data with $\text{Poisson}(1)$ random variables and fitting AEE. We bootstrap 1,000 times, and plot the sample mean of our learned mean functions under complete data. We also set counts to zero in intervals with missing counts, which biases the mean function estimates. Figure 2a compares inference from complete vs partially missing data using our wrapper vs zeroing out missing data. Our wrapper learns a model much closer to the complete data than its initialization or the zeroing model. Section D of the supplementary material has additional experiments: we investigate sensitivity to initialization and different missingness probabilities, replace AEE with other M-step methods, investigate missing at random (MAR) and missing not at random (MNAR), and show results for the baselines discussed above.

### 6.2 Smoking Cessation Study

We analyze data from an ongoing smoking cessation study in which the following EMA question was delivered randomly 3-4 times per day: "*Since the last assessment, how many cigarettes did you smoke?*" We have 125 participants tracked over 14 days, with 3-4 random EMAs targeted. The first

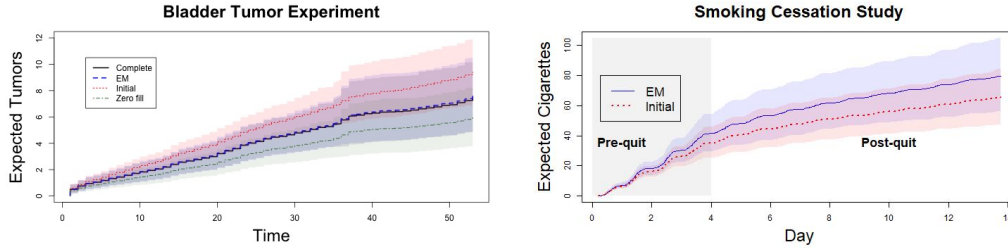

Figure 2: The estimated mean function and the 95% bootstrap confidence band from **a)** bladder tumor dataset: synthetic 20% missingness probability. Initialization has missing data set to Poisson(1) values. **b)** smoking cessation study. Initialization treats self-reported counts in intervals over 24 hours (about 5.6% of observations) as valid.

four days are their normal smoking behavior, while the remaining days involve them attempting to quit. In both curves the smoking rates are much higher pre-quit than post-quit, suggesting that they do in fact attempt to quit. After discussion with psychologists, we treat counts in intervals of longer than 24 hours as missing because they are considered to be unreliable. We use the counts to initialize our model. Unlike the previous experiments, we lack a ground truth. Note that this study is currently ongoing with more participants being added and we cannot share the dataset. However, a different subset of questions (as well as wearable sensors) was also used in [6], where they provide further description of the study.

Figure 2b shows the results based on $1,000$ non-parametric bootstrap samples and two models: one treating long intervals as valid ("initial"), and the other treating them as missing where we use EM. Long intervals make up 5.6% of observations. The EM algorithm estimates that on average smokers attempting to quit smoked $80.4$ cigarettes by the end of the study; in contrast, AEE underestimated ($66.11$), a difference of 21.7%. This is consistent with our collaborating behavioral scientists' hypothesis that participants may under-report a cigarette count when the gap between completed EMAs is large. The proposed functional EM is able to borrow count information across multiple shorter overlapping intervals from other participants to alleviate this under-reporting. We discuss other analyses of the dataset further in section E of the supplementary material.

## 7   Discussion

This paper proposed a functional EM algorithm to estimate the mean function for incomplete panel count data. Extending existing EM algorithm analysis to general non-parametric settings, we provided finite sample convergence guarantees to the truth. We conducted extensive experiments to illustrate the effectiveness of the proposed algorithm in recovering the true mean function. We applied the proposed algorithm to a smoking cessation study and found that participants may underestimate their cigarettes smoked over intervals longer than 24 hours. To the best of our knowledge, we are the first to apply panel count data methods to EMAs, despite their central role in behavioral science research.

The main theoretical limitation is the MCAR assumption. Deviations from MCAR can happen if non-reporting depends on a subject's emotional state, which may influence smoking counts. Extensions to other missingness mechanisms such as missing at random (MAR) warrant future research; our analysis provides a general framework and first step, which can eventually be extended to also incorporate covariates under a more complex missing data model. Like existing theoretical analysis of EM algorithms [34], another limitation is the need to optimize over a ball around the true mean function, which is unknown. Relaxing this condition is likely very challenging but also important future work. Finally, we would like test statistics for our estimator. This is challenging as we are not aware of any results showing that the difference between the true mean function and its estimator converges to a Gaussian process. However, there are test statistics based on weighting [2] that could be extended to this setting.

## Broader Impact

Understanding the dynamics for individuals who attempt to change and maintain behaviors to improve health has important societal value, for example, a comprehensive understanding of how smokers attempt to quit smoking may guide behavioral scientists to design better intervention strategies that tailor to the highest risk windows of relapse. Our theory and method provide an approach to understanding a particular aspect of the smoking behavior (mean function). The resulting algoithm is robust to Poisson process violations, readily adaptable and simple to implement, highlighting the potential for its wider adoption. The negative use case could be lack of sensitivity analysis around the assumptions such as missing data mechanism which may lead to misleading conclusions. Our current recommendation is to consult scientists about the plausibility of the assumption about missing data.

## 8 Funding and Other Acknowledgments

We acknowledge funding from the national institute of health (NIH) under awards U01CA229437, R01CA224537, R01MD010362, R01CA190329, R01 MH101459, R01 DA039901 as well as U54EB020404 (by NBIB) through funds provided by the Big Data to Knowledge (BD2K) initiative. We also acknowledge an investigator grant from Precision Health Initiative at the University of Michigan, Ann Arbor.

We thank Soujanya Chatterjee, Shahin Samiei, Timothy Hnat and Santosh Kumar at the University of Memphis for helpful discussions.

## Footnotes

[1] In principle a participant who completes very few EMAs but reports their counts with 100% accuracy is not creating missing data, because standard mean function estimators [31] would still be asymmptotically unbiased. In practice, however, behavioral scientists frequently treat EMAs as if they are missing if the interval to the last EMA exceeds a cut-off, such as 24 hours, due to recall bias.

[2]This is not strong, and doesn't imply at least a constant first derivative/superlinear function in general. Consider $\Lambda^*(t) = t^{1/2}$ over $[0, T_{\text{end}}]$. The derivative $\frac{1}{2\sqrt{t}} \geq \frac{1}{2\sqrt{T_{\text{end}}}}$, but this function is sub-linear.

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
