[Supplementary Material]

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

# A    Proofs Related to Theorem 1

## A.1    Proof of Equality in Equation 1

We start with the numerator

$$Q(\Lambda^{(l)} + \zeta\Lambda'|\Lambda) - Q(\Lambda^{(l)}|\Lambda) = P(\sum_{j=1}^{K} \Delta N_j^{\tau_j} \Delta\Lambda_j^{s_j} \log\left(\Delta\Lambda_j^{(l)} + \zeta\Delta\Lambda_j'\right) - (\Lambda_K^l + \zeta\Lambda_K'))$$

$$- P(\sum_{j=1}^{K} \Delta N_j^{\tau_j} \Delta\Lambda_j^{s_j} \log\left(\Delta\Lambda_j^{(l)}\right) - (\Lambda_K^l)))$$

$$= P(\sum_{j=1}^{K} \Delta N_j^{\tau_j} \Delta\Lambda_j^{s_j} \log \frac{\Delta\Lambda_j^{(l)} + \zeta\Delta\Lambda_j'}{\Delta\Lambda_j^{(l)}} - \zeta\Lambda_K')$$

Now consider

$$\lim_{\zeta\downarrow 0} \frac{\log \frac{\Delta\Lambda_j^{(l)} + \zeta\Delta\Lambda_j'}{\Delta\Lambda_j^{(l)}}}{\zeta} = \lim_{\zeta\downarrow 0} \log \left(\frac{\Delta\Lambda_j^{(l)} + \zeta\Delta\Lambda_j'}{\Delta\Lambda_j^{(l)}}\right)^{1/\zeta}$$

$$= \lim_{\zeta\downarrow 0} \log \left(1 + \zeta\frac{\Delta\Lambda_j'}{\Delta\Lambda_j^{(l)}}\right)^{1/\zeta}$$

$$= \log \exp\left(\frac{\Delta\Lambda_j'}{\Delta\Lambda_j^{(l)}}\right)$$

$$= \frac{\Delta\Lambda_j'}{\Delta\Lambda_j^{(l)}}$$

We next need to show that we can pull the limit as $\zeta \downarrow 0$ under integrals. There are two relevant terms: $\frac{P(\sum_{j=1}^{K} \zeta\Lambda_K')}{\zeta}$, which we can trivially handle by pulling $\zeta$ outside the integral, and

$$\frac{P(\sum_{j=1}^{K} \Delta N_j^{\tau_j} \Delta\Lambda_j^{s_j} \log \frac{\Delta\Lambda_j^{(l)} + \zeta\Delta\Lambda_j'}{\Delta\Lambda_j^{(l)}})}{\zeta} = P(\sum_{j=1}^{K} \Delta N_j^{\tau_j} \Delta\Lambda_j^{s_j} \frac{1}{\zeta} \log \frac{\Delta\Lambda_j^{(l)} + \zeta\Delta\Lambda_j'}{\Delta\Lambda_j^{(l)}})$$

Noting that $\frac{1}{\zeta} \log \frac{\Delta\Lambda_j^{(l)} + \zeta\Delta\Lambda_j'}{\Delta\Lambda_j^{(l)}}$ is monotone increasing to $\frac{\Delta\Lambda_j'}{\Delta\Lambda_j^{(l)}}$ as $\zeta \downarrow 0$, we can apply the monotone convergence theorem to pull the limit under the integral. Then

$$\lim_{\zeta\downarrow 0} \frac{Q(\Lambda^{(l)} + \zeta\Lambda'|\Lambda) - Q(\Lambda^{(l)}|\Lambda)}{\zeta} = P(\sum_{j=1}^{K} \Delta N_j^{\tau_j} \Delta\Lambda_j^{s_j} \frac{\Delta\Lambda_j'}{\Delta\Lambda_j^{(l)}} - \Lambda_K')$$

$$= P(\sum_{j=1}^{K} (\frac{\Delta N_j^{\tau_j} \Delta\Lambda_j^{s_j}}{\Delta\Lambda_j^l} - 1)(\Delta\Lambda_j')) \qquad (2)$$

## A.2    Proof of Lemma 1

$$\left|\sum_{j=1}^{K} s_j \left(\frac{\Delta\Lambda_j - \Delta\Lambda_j^*}{\Delta\Lambda_j^*}\right)(\Delta\Lambda_j' - \Delta\Lambda_j^*)\right| \leq k_0 \frac{\sup |(\Delta\Lambda - \Delta\Lambda^*)(\Delta\Lambda' - \Delta\Lambda^*))|}{\inf \Delta\Lambda^*}$$

$$= k_0 \frac{b^2}{c}$$

$$< \infty$$

then any integral of this over $\mathcal{T} \times \mathcal{K} \times \mathcal{Z}$ will also be finite, and we can apply Fubini's theorem to such integrals. Then recalling that $\epsilon > 0$ is the MCAR probability (assumption 8),

$$\langle \nabla Q(\Lambda^*|\Lambda) - \nabla Q(\Lambda^*|\Lambda^*), \Lambda' - \Lambda^* \rangle = P\left[\sum_{j=1}^{K} s_j \left(\frac{\Delta\Lambda_j - \Delta\Lambda_j^*}{\Delta\Lambda_j^*}\right)(\Delta\Lambda_j' - \Delta\Lambda_j^*)\right]$$

$$\leq \epsilon \left[P\left(\sum_{j=1}^{K}\left(\frac{\Delta\Lambda_j - \Delta\Lambda_j^*}{\Delta\Lambda_j^*}\right)^2\right)\right]^{1/2}$$

$$\times \left[P\left(\sum_{j=1}^{K}\left(\Delta\Lambda_j' - \Delta\Lambda_j^*\right)^2\right)\right]^{1/2}$$

Here we used Fubini's theorem to pull out $\epsilon$. Applying the CS inequality for inner products in $l^2$, and finally applying the CS inequality for expectations gives us the result. The first term on rhs is then:

$$\left[P\left(\sum_{j=1}^{K}\left(\frac{\Delta\Lambda_j - \Delta\Lambda_j^*}{\Delta\Lambda_j^*}\right)^2\right)\right]^{1/2} \leq \frac{1}{c}\left[P\left(\sum_{j=1}^{K}\left(\Delta\Lambda_j - \Delta\Lambda_j^*\right)^2\right)\right]^{1/2}$$

This gives us

$$P\left[\sum_{j=1}^{K} s_j \left(\frac{\Delta\Lambda_j - \Delta\Lambda_j^*}{\Delta\Lambda_j^*}\right)(\Delta\Lambda_j' - \Delta\Lambda_j^*)\right] \leq \frac{\epsilon}{c}\|\Lambda - \Lambda^*\|\|\Lambda' - \Lambda^*\|$$

and thus

$$\langle \nabla Q(\Lambda^*|\Lambda) - \nabla Q(\Lambda^*|\Lambda^*), \Lambda' - \Lambda^* \rangle \leq \frac{\epsilon}{c}\|\Lambda - \Lambda^*\|\|\Lambda' - \Lambda^*\|$$

### A.3 Proof of Lemma 2

Note that

$$Q(\Lambda'|\Lambda) = P\left(\sum_{j=1}^{K} \Delta N_j^{\tau_j}\Delta\Lambda_j^{s_j}\log[\Delta\Lambda_j'] - \Lambda'(T_K)\right)$$

$$= P\left(\sum_{j=1}^{K}\tau_j[\Delta N_j\log[\Delta\Lambda_j'] - \Lambda'(T_K)]\right) + P\left(\sum_{j=1}^{K}s_j[\Delta\Lambda_j\log[\Delta\Lambda_j'] - \Lambda'(T_K)]\right)$$

Define

$$Q_1(\Lambda'|\Lambda) = P\left(\sum_{j=1}^{K}\Delta N_j\log[\Delta\Lambda_j'] - \Lambda'(T_K)\right)$$

$$Q_2(\Lambda'|\Lambda) = P\left(\sum_{j=1}^{K}\Delta\Lambda_j\log[\Delta\Lambda_j'] - \Lambda'(T_K)\right)$$

and note that if $\sum_{j=1}^{K}\tau_j[\Delta N_j\log[\Delta\Lambda_j'] - \Lambda'(T_j)]$ and $\sum_{j=1}^{K}s_j[\Delta N_j\log[\Delta\Lambda_j'] - \Lambda'(T_j)]$ are integrable over $\mathcal{N} \times \mathcal{T} \times \mathcal{K} \times \mathcal{Z}$ with respect to the measure $P$, we can apply Fubini's theorem to obtain

$$Q(\Lambda'|\Lambda) = (1 - \epsilon)Q_1(\Lambda'|\Lambda) + \epsilon Q_2(\Lambda'|\Lambda)$$

This proof takes place in four parts. We first show an inequality that allows us to characterize $r$ for $B_r(\Lambda^*)$. Next we show that in this ball the integrability conditions above hold. We then show that $Q_1$ and $Q_2$ are both strictly concave. Finally we show that $Q_1$ is strongly concave and thus $Q$, a positive linear combination of a strongly concave and a strictly concave function, is strongly concave.

### A.3.1 Characterizing the Radius of Contraction

**Claim 1.** *Let* $h(x) = x(\log(x) - 1) + 1$. *For* $\|\Lambda^* - \Lambda'\|_\infty \leq \frac{c}{4}$

$$h\left(\frac{\Delta\Lambda_j^*}{\Delta\Lambda_j'}\right) \geq \frac{1}{3}\left(\frac{\Delta\Lambda_j^*}{\Delta\Lambda_j'} - 1\right)^2$$

*Proof.* Let $\phi(y) = (1 + y)(\log(1 + y) - 1) + 1$. Then we can Taylor expand

$$\phi(y) = \phi(0) + \phi'(0)y + \frac{\phi''(\xi)}{2}y^2$$

where $\xi$ lies between 0 and $y$. Then noting that $\phi(0) = \phi'(0) = 0$ and $\phi''(y) = \frac{1}{1+y}$,

$$\phi(y) = \frac{\phi''(\xi)}{2}y^2$$
$$= \frac{1}{2(1+\xi)}y^2$$
$$\geq \frac{1}{3}y^2$$

for $y \in (-1, 1)$. Letting $h(x) = x(\log x - 1) + 1$ this implies that $h(x) \geq \frac{1}{3}(x - 1)^2$ for $(x - 1) \in (-1, 1)$ i.e. $x \in (0, 2)$. Let $x = \frac{\Delta\Lambda_j^*}{\Delta\Lambda_j'}$. Then we need $\Delta\Lambda_j' \geq \frac{1}{2}\Delta\Lambda_j^*$ for $h(x) \geq \frac{1}{3}(x - 1)^2$ to hold. Note that if

$$|\Delta\Lambda_j^* - \Delta\Lambda_j'| \leq z\Delta\Lambda_j^* \tag{3}$$

Then

$$\Delta\Lambda_j^* - \Delta\Lambda_j' \leq z\Delta\Lambda_j^*$$
$$(1 - z)\Delta\Lambda_j^* \leq \Delta\Lambda_j'$$
$$\Delta\Lambda_j^* \leq \frac{1}{1 - z}\Delta\Lambda_j'$$

and if $z = 0.5$ the desired result holds. For equation 3 to hold it suffices to have $(\Delta\Lambda_j^* - \Delta\Lambda_j')^2 \leq z^2\Delta\Lambda_j^{*2} \leq z^2 c^2$. Now noting that $(a - b)^2 \leq 2(a^2 + b^2)$,

$$(\Delta\Lambda_j^* - \Delta\Lambda_j')^2 = ([\Lambda_j^* - \Lambda_j'] - [\Lambda_{j-1}^* - \Lambda_{j-1}'])^2$$
$$\leq 2([\Lambda_j^* - \Lambda_j']^2 + [\Lambda_{j-1}^* - \Lambda_{j-1}']^2)$$
$$\leq 4\|\Lambda^* - \Lambda'\|_\infty^2$$

w.p. 1, and noting that we want $z = 0.5$ a sufficient condition is

$$4\|\Lambda^* - \Lambda'\|_\infty^2 \leq z^2 c^2$$
$$\|\Lambda^* - \Lambda'\|_\infty \leq \frac{c}{4}$$

$\square$

### A.4 Integrability Conditions

First note by assumption 7, all moments of $N(T_{\text{end}})$ are uniformly bounded and thus $\|N(T_{\text{end}})\|_\infty < \infty$. Then for $\Lambda, \Lambda' \in B_r(\Lambda^*)$ and by assumption 3,

$$P\left|\sum_{j=1}^{K} \tau_j[\Delta N_j \log[\Delta\Lambda_j'] - \Lambda'(T_j)]\right| \leq k_0\|N(T_{\text{end}})\|_\infty \max(|\log\frac{c}{2}|, \log b) + k_0 b$$

$$< \infty$$

$$P\left|\sum_{j=1}^{K} s_j[\Delta\Lambda_j \log[\Delta\Lambda_j'] - \Lambda'(T_j)]\right| \leq k_0 b \max(|\log\frac{c}{2}|, \log b) + k_0 b$$

$$< \infty$$

### A.4.1 Strict Concavity of $Q_1$ and $Q_2$

For $Q_1$ we have

$$Q_1(\frac{\Lambda_1 + \Lambda_2}{2}|\Lambda) = P\left(\sum_{j=1}^{K} \Delta N_j \log[\frac{\Delta\Lambda_{1,j} + \Delta\Lambda_{2,j}}{2}] - \frac{\Lambda_{1,K} + \Lambda_{2,K}}{2}\right)$$

$$> P\left(\sum_{j=1}^{K} \Delta N_j \log[\sqrt{\Delta\Lambda_{1,j}}\sqrt{\Delta\Lambda_{2,j}}] - \frac{\Lambda_{1,K} + \Lambda_{2,K}}{2}\right) \quad \text{AM-GM inequality}$$

$$= \frac{1}{2}Q_1(\Lambda_1|\Lambda) + \frac{1}{2}Q_1(\Lambda_2|\Lambda)$$

and for $Q_2$ the same argument can be made.

### A.4.2 Strong Concavity of $Q$

Now note that

$$Q_1(\Lambda^*|\Lambda) - Q_1(\Lambda'|\Lambda) + \langle \nabla Q_1(\Lambda^*|\Lambda), \Lambda' - \Lambda^* \rangle$$

$$= P\left(\sum_{j=1}^{K}[\Delta N_j \log \frac{\Delta\Lambda_j^*}{\Delta\Lambda_j'} - (\Delta\Lambda_j^* - \Delta\Lambda_j')(1 - (\frac{\Delta N_j}{\Delta\Lambda_j^*} - 1))]\right)$$

$$= P\left(\sum_{j=1}^{K}[\Delta\Lambda_j^* \log[\frac{\Delta\Lambda_j^*}{\Delta\Lambda_j'}] - (\Delta\Lambda_j^* - \Delta\Lambda_j')]\right)$$

$$= P\left(\sum_{j=1}^{K} \Delta\Lambda_j'\left(\frac{\Delta\Lambda_j^*}{\Delta\Lambda_j'} \log \frac{\Delta\Lambda_j^*}{\Delta\Lambda_j'} - (\frac{\Delta\Lambda_j^*}{\Delta\Lambda_j'} - 1)\right)\right)$$

$$= P\left(\sum_{j=1}^{K} \Delta\Lambda_j' h(\frac{\Delta\Lambda^*}{\Delta\Lambda_j'})\right)$$

$$\geq \frac{1}{3}P\left(\sum_{j=1}^{K} \Delta\Lambda_j'(\frac{\Delta\Lambda_j^*}{\Delta\Lambda_j'} - 1)^2\right)$$

$$\geq \frac{1}{3b}\|\Lambda' - \Lambda^*\|^2$$

$$\geq \frac{1}{3b}\|\Lambda' - \Lambda^*\|^2$$

where the first inequality holds by claim 1. Now by strict concavity of $Q_2$ as shown in A.4.1 we have

$$Q_2(\Lambda^*|\Lambda) - Q_2(\Lambda'|\Lambda) + \langle \nabla Q_2(\Lambda^*|\Lambda), \Lambda' - \Lambda^* \rangle \geq 0$$

Summing $(1 - \epsilon)Q_1$ and $\epsilon Q_2$ we obtain

$$Q(\Lambda^*|\Lambda) - Q(\Lambda'|\Lambda) + \langle \nabla Q(\Lambda^*|\Lambda), \Lambda' - \Lambda^* \rangle \geq \frac{(1 - \epsilon)}{3b}\|\Lambda' - \Lambda^*\|^2$$

### A.5 Proof of Population Contractivity

We now state the main proof of population contractivity. Denote

$$V(\Lambda'|\Lambda) = Q(\Lambda'|\Lambda) - Q(\Lambda^*|\Lambda) - \langle \nabla Q(\Lambda^*|\Lambda), \Lambda' - \Lambda^* \rangle$$

then

$$0 \leq Q(\Lambda'|\Lambda) - Q(\Lambda^*|\Lambda)$$
$$= V(\Lambda'|\Lambda) + \langle \nabla Q(\Lambda^*|\Lambda), \Lambda' - \Lambda^* \rangle$$
$$= V(\Lambda'|\Lambda) + \langle \nabla Q(\Lambda^*|\Lambda) - \nabla Q(\Lambda^*|\Lambda^*), \Lambda' - \Lambda^* \rangle + \langle \nabla Q(\Lambda^*|\Lambda^*), \Lambda' - \Lambda^* \rangle$$
$$\leq V(\Lambda'|\Lambda) + \langle \nabla Q(\Lambda^*|\Lambda) - \nabla Q(\Lambda^*|\Lambda^*), \Lambda' - \Lambda^* \rangle \text{ KKT conditions}$$
$$\leq -\nu\|\Lambda' - \Lambda^*\|^2 + \gamma\|\Lambda - \Lambda^*\|\|\Lambda' - \Lambda^*\| \text{ technical Lemmas}$$

and rearranging terms and dividing both sides by $\|\Lambda' - \Lambda^*\|$ gives the desired result. Note that we used $\langle \nabla Q(\Lambda^*|\Lambda^*), \Lambda' - \Lambda^* \rangle \leq 0$, which if $\langle \cdot, \cdot \rangle$ were a valid inner product would be the KKT conditions. However since $\langle \cdot, \cdot \rangle$ may not be a valid inner product, they must be checked specifically. [31] does it in the sample case for the true log-likelihood: it is easy to verify that it still holds in the population case for $Q$-functionals. Noting that $\Lambda^*$ maximizes $Q(\Lambda'|\Lambda^*)$, we have that $Q(\Lambda^* + \zeta(\Lambda' - \Lambda^*)|\Lambda^*) - Q(\Lambda^*|\Lambda^*) \leq 0$

$$\langle \nabla Q(\Lambda^*|\Lambda^*), \Lambda' - \Lambda^* \rangle = \lim_{\zeta \downarrow 0} \frac{Q(\Lambda^* + \zeta(\Lambda' - \Lambda^*)|\Lambda^*) - Q(\Lambda^*|\Lambda*)}{\zeta}$$
$$\leq 0$$

# B Proofs Related to Theorem 2

## B.1 Proof of Proposition 2

### B.1.1 Definitions and Background from the Literature

Before proving the proposition, we restate several important results that we use from existing literature. We repeat two definitions and two theorems from [25], adjusted to our notation. Note that $\eta$-brackets are normally called $\epsilon$-brackets. However, since we have already used $\epsilon$ to denote MCAR probabilities, we call them $\eta$-brackets.

**Definition 11.** *($\eta$-bracket) Let $(\mathcal{F}, d)$ be a normed space of functions with distance metric $d$ induced by some norm. Given two functions $l(\cdot)$ and $g(\cdot)$, the bracket $[l, g]$ is the set of all functions $f \in \mathcal{F}$ with $l(u) \leq f(u) \leq g(u) \forall u \in [0, T_{end}]$. An $\eta$-bracket is a bracket $[l, g]$ with $d(l, g) < \eta$.*

**Definition 12.** *(Bracketing numbers). The bracketing number $N_{[]}(\eta, \mathcal{F}, L_2(P))$ is the minimum number of $\eta$-brackets needed to cover $\mathcal{F}$ using $L^2(P)$ distance.*

**Definition 13.** *(Bracketing Integral) The bracketing integral is defined as*

$$J_{[]}(\delta, \mathcal{F}, L_2(P)) \equiv \int_0^\delta \sqrt{\log\big(N_{[]}(\eta, \mathcal{F} \cup \{0\}, L_2(P))\big)} d\eta$$

Note that since any non-empty set requires at least one bracket to cover it and $\log(x + 1) \leq 1 + \log(x)$ for $x \geq 1$,

$$\int_0^\delta \sqrt{\log\big(N_{[]}(\eta, \mathcal{F} \cup \{0\}, L_2(P))\big)} \leq \int_0^\delta \sqrt{1 + \log\big(N_{[]}(\eta, \mathcal{F}, L_2(P))\big)}$$

the right hand side is sometimes used as the definition of the bracketing integral, but we use the left hand side, following [25]. We now restate a theorem from [25], with the notation heavily adapted to our setting for clarity. The theorem is otherwise the same.

**Theorem 3.** *(Theorem 5.1 in [25]) Let $(\Theta \cap B_r(\Lambda^*), \|\cdot\|)$ be a semi-metric space. Fix $n \geq 1$. Let $\{Q_n(\Lambda'|\Lambda) : \Lambda' \in \Theta \cap B_r(\Lambda^*)\}$ be a stochastic process and $\{Q(\Lambda'|\Lambda) : \Lambda' \in \Theta \cap B_r(\Lambda^*)\}$ be a deterministic process. Assume*

$$Q(\Lambda'|\Lambda) - Q(M(\Lambda)|\Lambda) \leq -c_1\|\Lambda' - M(\Lambda)\|^2 \qquad (4)$$

*for some $c_1 > 0$. We call this the separation condition. Further, let*

$$U_n(\Lambda'|\Lambda) = Q_n(\Lambda'|\Lambda) - Q(\Lambda'|\Lambda)$$

*and assume that there exists some function $\phi_n(\cdot)$ satisfying the following three conditions*

1. *The following expected supremum condition holds*

$$\mathbb{E}\left[\sup_{\Lambda':\|\Lambda'-M(\Lambda)\|\leq\delta}\sqrt{n}|U_n(\Lambda'|\Lambda)-U_n(M(\Lambda)|\Lambda)|\right]\lesssim\phi_n(\delta) \tag{5}$$

2. *there exists $\alpha < 2$ so that*

$$\phi_n(dx)\leq d^\alpha\phi_n(x)\forall d>1,x>0 \tag{6}$$

3. *for the rate of convergence $r_n$*

$$\phi_n(r_n)\lesssim\sqrt{n}r_n^2 \tag{7}$$

   *as $n$ varies.*

*Here $\lesssim$ means $\leq$ the right hand side times a constant. Then for every $L > 0$, $\|M_n(\Lambda) - M(\Lambda)\| \leq 2^L r_n$ with probability at least $1 - u_L$. Here $u_L = \tilde{c}\sum_{j>M} 2^{j(\alpha-2)}$, where $\tilde{c}$ only depends on the constants in the separation condition and the expected supremum bound.*

Note that this is essentially a special case of Theorem 3.2.5 of [27], but it uses expectations instead of outer expectations. It makes the stronger version of their assumptions and draws a stronger conclusion, giving a finite sample bound. The key in our setting will be to ensure that $\tilde{c}$ does not vary across iterations, which requires that the constants for the separation condition and the expected supremum bound do not vary across iterations. Importantly, we cannot always apply this theorem in the general functional EM setting: it requires that the sample $Q$-functional at its maximizer over a Sieve is equal to the sample $Q$-functional at its maximizer over the full function space. This holds for the Poisson process log-likelihood for panel count data for a range of cases: for instance with step functions or splines where jumps/knots occur at observation times. It does not necessarily hold for arbitrary models. In the latter case we may be able to use Theorem 6.1 of [25], but this would require checking it carefully for each potential model. We also note

**Theorem 4.** *(Theorem 4.12 of [25]) For any class $\mathcal{F}$ of measurable functions $f : \mathcal{X} \to \mathbb{R}$ such that $Pf^2 < \delta^2$ and $\|f\|_\infty \leq M$ for every $f$,*

$$\mathbb{E}[\sup_{f\in\mathcal{F}}|\sqrt{n}(P_n-P)|]\leq\tilde{K}J_{[]}(\delta,\mathcal{F},L_2(P))\left(1+\frac{J_{[]}(\delta,\mathcal{F},L_2(P))}{\delta^2\sqrt{n}}M\right)$$

*where $\tilde{K} > 0$ is some constant.*

Importantly, $\tilde{K}$ is a *universal* constant and does not depend on $\mathcal{F}$. This was noted by [20]. A version of this theorem was originally Lemma 3.4.2 in [27].

### B.1.2 Outline

We follow [1], which proves the rate of convergence for the maximum pseudo-likelihood of a Poisson process objective function for panel count data: extending their proof to the expected complete data log-likelihood case is straightforward. However, we face the issue that we want a high probability uniform bound on the distance between sample and population M-steps *across* EM iterations, whereas they neeeded an asymptotic high probability rate of convergence for the pseudo MLE. This poses three challenges: 1) our objective function at each iteration is the expected log-likelihood rather than the log-likelihood. Thus we cannot prove that the separation condition holds using the same techniques. 2) the separation condition must hold always rather than only in a neighborhood of the optimum 3) we need to check that constants are the same across EM iterations. Our aim is to apply Theorem 3 and show that the constant $\tilde{c}$ in $u_L$ does not vary across EM iterations. Note that other than checking the separation condition, the majority of this proof simply repeats the proof strategy of [1] but fills in details of results they call to make sure that constants don't vary across iterations of EM.

This proof takes place in four parts. We first show that the separation condition holds. We then bound the expectation of the supremum of the magnitude of an empirical process. We next prove the two properties of the function involved in that bound to show the rate of convergence, and finally conclude by applying Theorem 3.

### B.1.3 Separation Condition

We first prove that the separation condition given by Equation 4 holds. This involves applying functional second order Taylor expansions to $Q_1$ and $Q_2$ and using the remainder terms to obtain quadratic lower bounds.

**Claim 2.** *For any $\Lambda, \Lambda' \in B_r(\Lambda^*)$,*

$$Q(M(\Lambda)|\Lambda) - Q(\Lambda'|\Lambda) \geq \left( (1-\epsilon)\frac{c}{b^2} + \epsilon\frac{c}{2b^2} \right) \|M(\Lambda) - \Lambda'\|^2$$

*Proof.* Consider the functional second order expansion of $Q_1(\Lambda'|\Lambda)$ and $Q_2(\Lambda'|\Lambda)$ at $M(\Lambda)$. Note that this is valid by considering $\Lambda' = M(\Lambda) + \zeta(\Lambda' - M(\Lambda))$ for $\zeta = 1$. We can then consider $g_1(\zeta) = Q(M(\Lambda) + \zeta(\Lambda' - M(\Lambda))|\Lambda)$ for fixed $\Lambda, M(\Lambda), \Lambda'$ and similarly for $g_2$ and do a second order Taylor expansion at $\zeta = 0$ as (See [11] for further details)

$$g_1(1) = g_1(0) + g_1'(0) + \frac{1}{2}g_1''(\zeta)$$

for some $\zeta_1 \in (0,1)$. We first assume that $g_1$ is twice differentiable in the interval $[0,1)$, and then show it rigorously. We then have

$$
\begin{aligned}
Q_1(M(\Lambda)|\Lambda) - Q_1(\Lambda'|\Lambda) &= -\langle \nabla Q_1(M(\Lambda)|\Lambda), \Lambda' - M(\Lambda) \rangle \\
&\quad - P\Big(-\sum_{j=1}^{K} (\Delta M(\Lambda)_j - \Delta\Lambda_j')\frac{\Delta N_j}{\Delta\xi_{1,j}^2}(\Delta M(\Lambda)_j - \Delta\Lambda_j')\Big) \\
&= -\langle \nabla Q_1(M(\Lambda)|\Lambda), \Lambda' - M(\Lambda) \rangle \\
&\quad + P\Big(\sum_{j=1}^{K} (\Delta M(\Lambda)_j - \Delta\Lambda_j')\frac{\Delta\Lambda_j^*}{\Delta\xi_{1,j}^2}(\Delta M(\Lambda)_j - \Delta\Lambda_j')\Big) \\
&\geq -\langle \nabla Q_1(M(\Lambda)|\Lambda), \Lambda' - M(\Lambda) \rangle + \frac{c}{b^2}\|M(\Lambda) - \Lambda'\|^2
\end{aligned}
$$

here $\xi_{1,j} = M(\Lambda) + \zeta_1(\Lambda' - M(\Lambda))$. Similarly for $g_2$ and $Q_2$,

$$
\begin{aligned}
Q_2(M(\Lambda)|\Lambda) - Q_2(\Lambda'|\Lambda) &= -\langle \nabla Q_2(M(\Lambda)|\Lambda), \Lambda' - M(\Lambda) \rangle \\
&\quad - P\Big(-\sum_{j=1}^{K} (\Delta M(\Lambda)_j - \Delta\Lambda_j')\frac{\Delta\Lambda_j}{\Delta\xi_{2,j}^2}(\Delta M(\Lambda)_j - \Delta\Lambda_j')\Big) \\
&= -\langle \nabla Q_2(M(\Lambda)|\Lambda), \Lambda' - M(\Lambda) \rangle \\
&\quad + P\Big(\sum_{j=1}^{K} (\Delta M(\Lambda)_j - \Delta\Lambda_j')\frac{\Delta\Lambda_j}{\Delta\xi_{2,j}^2}(\Delta M(\Lambda) - \Delta\Lambda')\Big) \\
&\geq -\langle \nabla Q_2(M(\Lambda)|\Lambda), \Lambda' - M(\Lambda) \rangle + \frac{c}{2b^2}\|M(\Lambda) - \Lambda'\|^2
\end{aligned}
$$

where the last line follows since $\Lambda \in B_r(\Lambda^*)$ so that $\|\Lambda - \Lambda^*\|_\infty \leq \frac{c}{4}$ and thus $\Delta\Lambda \geq \frac{1}{2}\Delta\Lambda^* \geq \frac{1}{2}c$ w.p. 1. Noting that $\langle \nabla Q(M(\Lambda)|\Lambda), \Lambda' - M(\Lambda) \rangle \leq 0$ by the KKT conditions,

$$Q(M(\Lambda)|\Lambda) - Q(\Lambda'|\Lambda) \geq \left( (1-\epsilon)\frac{c}{b^2} + \epsilon\frac{c}{2b^2} \right) \|M(\Lambda) - \Lambda'\|^2$$

Thus the separation condition holds since we optimized over $\Theta \cap B_r(\Lambda^*)$.

We now return to showing that $g_1$ and $g_2$ are twice differentiable on $[0,1)$. Note that the derivatives of $g_1$ and $g_2$ at 0 are in fact the Gateaux derivatives of $Q_1$ and $Q_2$ in the direction of $\Lambda' - M(\Lambda)$. In A.1 of the supplementary material we previously showed the existence of right Gateaux derivatives. The change here is that we need to handle the fact that $\zeta$ may approach 0 from the left now. We first show that $g_1$ is differentiable at 0.

Recall

$$Q_1(M(\Lambda) + \zeta(\Lambda' - M(\Lambda))|\Lambda) = P(\sum_{j=1}^{K} \Delta N_j \log[\Delta M(\Lambda)_j + \zeta(\Delta\Lambda'_j - \Delta M(\Lambda)_j)]$$
$$- (M(\Lambda)(T_K) + \zeta(\Lambda'(T_K) - M(\Lambda)(T_K))))$$

so that

$$q'_1(0) = \lim_{\zeta \to 0} \frac{P(\sum_{j=1}^{K} \Delta N_j \log \frac{\Delta M(\Lambda)_j + \zeta(\Delta\Lambda'_j - \Delta M(\Lambda)_j)}{\Delta M(\Lambda)_j}) - \zeta(\Lambda'(T_K) - M(\Lambda)(T_K)))}{\zeta}$$

To show existence, we must show first that we can pull the derivative under the integral and second that we can differentiate the logarithm terms, which we can as long as the terms inside the logarithm are positive. For the first part, we show that we can bound the terms with $\log$ in the derivative, so that we can apply the bounded convergence theorem. The second part follows since in showing the first part we also show that the terms inside the logarithm are positive.

In order to apply the bounded convergence theorem we need to show that in a neighborhood of $\zeta = 0$, that

$$\frac{1}{\zeta} \log \frac{\Delta M(\Lambda)_j + \zeta(\Delta\Lambda'_j - \Delta M(\Lambda)_j)}{\Delta M(\Lambda)_j}$$

exists and is bounded. Since $1 - \frac{1}{x} \leq \log x \leq x - 1$ if $x > 0$ (we will show that $x > 0$ in this setting), we have

$$\log \frac{\Delta M(\Lambda)_j + \zeta(\Delta\Lambda'_j - \Delta M(\Lambda)_j)}{\Delta M(\Lambda)_j} \leq \zeta \frac{\Delta\Lambda'_j - \Delta M(\Lambda)_j}{\Delta M(\Lambda)_j}$$

$$\log \frac{\Delta M(\Lambda)_j + \zeta(\Delta\Lambda'_j - \Delta M(\Lambda)_j)}{\Delta M(\Lambda)_j} \geq \zeta \frac{\Delta\Lambda'_j - \Delta M(\Lambda)_j}{\Delta M(\Lambda)_j + \zeta(\Delta\Lambda'_j - \Delta M(\Lambda)_j)}$$

so that

$$\left| \frac{1}{\zeta} \log \frac{\Delta M(\Lambda)_j + \zeta(\Delta\Lambda'_j - \Delta M(\Lambda)_j)}{\Delta M(\Lambda)_j} \right| \leq \max \left( \left| \frac{\Delta\Lambda'_j - \Delta M(\Lambda)_j}{\Delta M(\Lambda)_j + \zeta(\Delta\Lambda'_j - \Delta M(\Lambda)_j)} \right|, \left| \frac{\Delta\Lambda'_j - \Delta M(\Lambda)_j}{\Delta M(\Lambda)_j} \right| \right)$$

Now $\Delta M(\Lambda)_j \geq \frac{c}{2}$ and we previously showed that $|\Delta\Lambda_j - \Delta\Lambda^*_j| \leq 2\|\Lambda - \Lambda^*\|_\infty \leq \frac{c}{2}$ (first inequality was proven in Claim 1, second by assumption) , so that

$$|\Delta\Lambda'_j - \Delta M(\Lambda)_j| \leq |\Delta\Lambda'_j - \Delta\Lambda^*_j| + |\Delta M(\Lambda)_j - \Delta\Lambda^*_j|$$
$$\leq c$$

thus if we consider $\zeta \in (-\frac{1}{4}, \frac{1}{4})$,

$$\Delta M(\Lambda)_j + \zeta(\Delta\Lambda'_j - \Delta M(\Lambda)_j) \geq \frac{c}{2} + \min(\zeta(\Delta\Lambda'_j - \Delta M(\Lambda)_j), -\zeta(\Delta\Lambda'_j - \Delta M(\Lambda)_j))$$
$$\geq \frac{c}{2} + \min(\zeta c, -\zeta c)$$
$$\geq \frac{c}{4}$$

giving us

$$\left| \frac{1}{\zeta} \log \frac{\Delta M(\Lambda)_j + \zeta(\Delta\Lambda'_j - \Delta M(\Lambda)_j)}{\Delta M(\Lambda)_j} \right| \leq \max \left( \frac{c}{\frac{c}{4}}, \frac{c}{\frac{c}{2}} \right)$$
$$= 4$$

We also needed to show that $\frac{\Delta M(\Lambda)_j + \zeta(\Delta\Lambda'_j - \Delta M(\Lambda)_j)}{\Delta M(\Lambda)_j} > 0$. However, we did this by showing that for $\zeta \in (-\frac{1}{4}, \frac{1}{4})$, the numerator is positive. Thus $g_1$ is differentiable at 0. Showing the existence of the second derivative is straightforward. For the derivative at points in $(0, 1)$ all terms are non-negative and thus it is simple to show existence using similar methods. Essentially the same argument can be applied to $g_2$ and $Q_2$.

$\square$

### B.1.4 Bounding the Expectation of the Supremum of the Magnitude of the Empirical Process

Our aim in this section is to apply Theorem 4 and use the result to show that the expected supremum condition, Equation 5 in Theorem 3 holds. Let

$$\Theta_\delta \equiv \{\Lambda' : \|\Lambda' - M(\Lambda)\| \leq \delta, \Lambda' \in \Theta \cap B_r(\Lambda^*)\}$$

Let $m_{\Lambda',\Lambda}(Y) \equiv \sum_{j=1}^K \Delta N_j^{\tau_j} \Delta \Lambda^{s_j}(T_j) \log[\Delta \Lambda'_j] - \Lambda'(T_K)$ and

$$\mathcal{M}_\delta \equiv \{m_{\Lambda',\Lambda}(Y) - m_{M(\Lambda),\Lambda}(Y) : \Lambda' \in \Theta_\delta\}$$

This section proceeds as follows. We first show that for all $f \in \mathcal{M}_\delta$, $Pf^2 \leq c_2 \delta^2$ for some constant $c_2 > 0$ and $\|f\|_\infty \leq c_3$ for some $c_3 > 0$. We next show a bound on the bracketing entropy in terms of the bracket size. We then use this to bound the bracketing integral using $\delta^{1/2}$. We combine all of this to bound the expectation of the supremum of interest. We must carefully note that relevant constants do not vary across iterations.

**Claim 3.** *For $\Lambda' \in \Theta_\delta$, $P|m_{\Lambda',\Lambda}(Y) - m_{M(\Lambda),\Lambda}(Y)|^2 \leq c_2 \delta^2$ for some $c_2 > 0$ that does not depend on $\Lambda'$ or $\Lambda$.*

*Proof.*

$$P(m_{\Lambda',\Lambda}(Y) - m_{M(\Lambda),\Lambda}(Y))^2$$

$$= P\left(\sum_{j=1}^K [\Delta N_j^{\tau_j} \Delta \Lambda_j^{s_j} \log \frac{\Delta \Lambda'_j}{\Delta M(\Lambda)_j} - (\Delta \Lambda'_j - \Delta M(\Lambda)_j)]\right)^2$$

$$\leq k_0 P\left(\sum_{j=1}^K [\Delta N_j^{\tau_j} \Delta \Lambda_j^{s_j} \log \frac{\Delta \Lambda'_j}{\Delta M(\Lambda)_j} - (\Delta \Lambda'_j - \Delta M(\Lambda)_j)]^2\right)$$

Cauchy Schwarz

$$\leq 2k_0 P\left(\sum_{j=1}^K [(\Delta N_j^{\tau_j} \Delta \Lambda_j^{s_j})^2 (\log \frac{\Delta \Lambda'_j}{\Delta M(\Lambda)_j})^2 + (\Delta \Lambda'_j - \Delta M(\Lambda)_j)^2]\right)$$

since $(a-b)^2 \leq 2(a^2 + b^2)$

$$= 2k_0 \left[P\left(\sum_{j=1}^K (\Delta N_j^{\tau_j} \Delta \Lambda_j^{s_j})^2 (\log \frac{\Delta \Lambda'_j}{\Delta M(\Lambda)_j})^2\right) + \|\Lambda' - M(\Lambda)\|^2\right] \qquad (8)$$

Then note

$$P\left(\sum_{j=1}^K (\Delta N_j^{\tau_j} \Delta \Lambda_j^{s_j} \log \frac{\Delta \Lambda'_j}{\Delta M(\Lambda)_j})^2\right) \leq P\left(\sum_{j=1}^K (\Delta N_j^{\tau_j} \Delta \Lambda_j^{s_j})^2 \frac{(\Delta \Lambda'_j - \Delta M(\Lambda)_j)^2}{\min(\Delta \Lambda'_j, \Delta M(\Lambda)_j)^2}\right)$$

$$\leq \frac{1}{4c^2} P\left(\sum_{j=1}^K (\Delta N_j^{\tau_j} \Delta \Lambda_j^{s_j})^2 (\Delta \Lambda'_j - \Delta M(\Lambda)_j)^2\right)$$

$$\leq \frac{\max(\|N(T_{\text{end}})\|_\infty^2, b^2)}{4c^2} P\left(\sum_{j=1}^K (\Delta \Lambda'_j - \Delta M(\Lambda)_j)^2\right)$$

assumption 7

$$= \frac{\max(\|N(T_{\text{end}})\|_\infty^2, b^2)}{4c^2} \|\Lambda' - M(\Lambda)\|^2$$

where the first line uses the inequality $1-\frac{1}{x} \le \log(x) \le x-1$ for $x > 0$ which implies $\left(\log\left(\frac{x}{y}\right)\right)^2 \le (x-y)^2/\min(x,y)^2$. Plugging this back into equation 8 we obtain

$$P(m_{\Lambda',\Lambda}(Y) - m_{M(\Lambda),\Lambda}(Y))^2 \le 2k_0 \left[\frac{\max(\|N(T_{\mathrm{end}})\|_\infty^2, b^2)}{4c^2}\|\Lambda' - M(\Lambda)\|^2 + \|\Lambda' - M(\Lambda)\|^2\right]$$

$$= \left(2k_0\frac{\max(\|N(T_{\mathrm{end}})\|_\infty^2, b^2)}{4c^2} + 1\right)\|\Lambda' - M(\Lambda)\|^2$$

$$\le \left(2k_0\frac{\max(\|N(T_{\mathrm{end}})\|_\infty^2, b^2)}{4c^2} + 1\right)\delta^2$$

so that we have $P|m_{\Lambda',\Lambda}(Y) - m_{M(\Lambda),\Lambda}(Y)|^2 \le c_2\delta^2$ for some constant $c_2$, and $c_2$ does not depend on either $\Lambda'$ or $\Lambda$. $\qquad\square$

**Claim 4.** *For $\Lambda' \in \Theta_\delta$, $\|m_{\Lambda',\Lambda}(Y) - m_{M(\Lambda),\Lambda}(Y)\|_\infty \le c_3$, where $c_3$ does not depend on $\Lambda$ or $\Lambda'$.*

*Proof.* Again using $\left|\log\left(\frac{x}{y}\right)\right| \le |x-y|/|\min(x,y)|$, we have

$$\left|m_{\Lambda',\Lambda}(Y) - m_{M(\Lambda),\Lambda}(Y)\right| = \left|\sum_{j=1}^{K}[\Delta N_j^{\tau_j}\Delta\Lambda_j^{s_j}\log\frac{\Delta\Lambda'_j}{\Delta M(\Lambda)_j} - (\Delta\Lambda'_j - \Delta M(\Lambda)_j)]\right|$$

$$\le \left|\sum_{j=1}^{K}\max(\|N(T_{\mathrm{end}})\|_\infty, b)\frac{\Delta\Lambda'_j - \Delta M(\Lambda)_j}{\min(\Delta\Lambda'_j, \Delta M(\Lambda)_j)}\right|$$

$$+ \left|\sum_{j=1}^{K}(\Delta\Lambda'_j - \Delta M(\Lambda)_j)\right|$$

$$\le k_0\frac{\max(\|N(T_{\mathrm{end}})\|_\infty, b)}{2c}[2\|\Lambda' - M(\Lambda)\|_\infty] + 2k_0\|\Lambda' - \Lambda^*\|_\infty$$

$$\le k_0\frac{\max(\|N(T_{\mathrm{end}})\|_\infty, b)}{2c}2[\|\Lambda' - \Lambda^*\|_\infty$$

$$+ \|M(\Lambda) - \Lambda^*\|_\infty] + 2k_0\|\Lambda' - \Lambda^*\|_\infty$$

$$\le k_0\frac{\max(\|N(T_{\mathrm{end}})\|_\infty, b)}{2} + k_0\frac{c}{2}$$

$$= c_3$$

where we used that $\Lambda', \Lambda^*, M(\Lambda) \in B_r(\Lambda^*)$ an $L^\infty$ ball with $r = \frac{c}{4}$ $\qquad\square$

By theorem 2.7.5 of [27], which bounds the bracketing number of monotone functions mapping to $[0,1]$, and noting that $\mathcal{M}_\delta$ has bracketing number less than or equal to that of $\Theta \cap B_r(\Lambda^*)$, which was shown in [2],

$$\log N_{[]}(\eta, \mathcal{M}_\delta, L_2(P)) \le c_4\eta^{-1}$$

where $c_4$ only depends on $U_{\mathrm{all}}$, the uniform upper bound in $\Theta$. Noting that $\mathcal{M}_\delta \cup \{0\} = \mathcal{M}_\delta$, we have

$$\int_0^\delta \sqrt{\log N_{[]}(\eta, \mathcal{M}_\delta \cup \{0\}, L_2(P))}d\eta \le c_5\delta^{1/2}$$

where again $c_5$ only depends on $U_{\mathrm{all}}$. Let

$$\|\sqrt{n}(P_n - P)\|_{M_\delta} = \sup_{f \in M_\delta}|\sqrt{n}(P_n - P)f|$$

and note that

$$\mathbb{E}\|\sqrt{n}(P_n - P)\|_{M_\delta} = \mathbb{E}\left[\sup_{\Lambda':\|\Lambda' - M(\Lambda)\| \le \delta}\sqrt{n}|U_n(\Lambda'|\Lambda) - U_n(M(\Lambda)|\Lambda)|\right]$$

**Claim 5.** *Because $P|m_{\Lambda',\Lambda}(Y) - m_{M(\Lambda),\Lambda}(Y)|^2 \leq c_2\delta^2$ and $\|m_{\Lambda',\Lambda}(Y) - m_{M(\Lambda),\Lambda}(Y)\|_\infty \leq c_3$, we have*

$$\mathbb{E}\|\sqrt{n}(P_n - P)\|_{\mathcal{M}_\delta} \leq c_6\phi_n(\delta)$$

*for $\phi_n(\delta) = \delta^{1/2} + \delta^{-1}n^{-1/2}$, where $c_6$ does not depend on $\Lambda'$ or $\Lambda$.*

*Proof.* Here we use Theorem 4 in order to prove Equation 5 in Theorem 3. By Theorem 4 and again noting that $\mathcal{M}_\delta \cup \{0\} = \mathcal{M}_\delta$, we have

$$\mathbb{E}\|\sqrt{n}(P_n - P)\|_{\mathcal{M}_\delta}$$

$$\leq \tilde{K}\left(\frac{(\int_0^{\sqrt{c_2}\delta}\sqrt{\log N_{[]}(\eta, \mathcal{M}_\delta, L_2(P))}d\eta)^2 c_3}{c_2\delta^2\sqrt{n}} + \int_0^{\sqrt{c_2}\delta}\sqrt{\log N_{[]}(\eta, \mathcal{M}_\delta, L_2(P))}d\eta\right)$$

$$\leq \tilde{K}\left(\frac{c_5^2(c_2^{1/2})c_3\delta}{c_2\delta^2\sqrt{n}} + c_5 c_2^{1/4}\delta^{1/2}\right)$$

$$= \tilde{K}\left(\frac{c_5^2 c_3}{c_2^{1/2}\delta\sqrt{n}} + c_5 c_2^{1/4}\delta^{1/2}\right)$$

$$\leq \tilde{K}\max\left(\frac{c_5^2 c_3}{c_2^{1/2}}, c_5 c_2^{1/4}\right)\left(\frac{1}{\delta\sqrt{n}} + \delta^{1/2}\right)$$

Set $c_6 = \tilde{K}\max\left(\frac{c_5^2 c_3}{c_2^{1/2}}, c_5 c_2^{1/4}\right)$ and we are done. $\square$

With this we have the bound on the expected supremum and have proven Equation 5 for Theorem 3.

### B.1.5   Characterizing the Function in the Bound

We first prove Equation 6 in Theorem 3.

$$\phi_n(d\delta) = (d\delta)^{1/2} + \frac{1}{(d\delta)\sqrt{n}}$$

$$= \sqrt{d}\left(\delta^{1/2} + \frac{1}{d^{3/2}\delta\sqrt{n}}\right)$$

$$\leq \sqrt{d}\left(\delta^{1/2} + \frac{1}{\delta\sqrt{n}}\right) \text{ for } d \geq 1$$

$$= \sqrt{d}\phi_n(\delta)$$

and thus $\alpha = \frac{1}{2}$.

Next we prove Equation 7. Let $r_n = n^{-1/3}$. Then

$$\phi_n(r_n) = 2n^{-1/6}$$

and

$$\sqrt{n}r_n^2 = \sqrt{n}(n^{-1/3})^2$$

$$= n^{-1/6}$$

and thus

$$\phi_n(r_n) \leq 2\sqrt{n}r_n^2$$

### B.1.6   Putting it All Together

We have now proven that all of the assumptions of Theorem 3 hold, and that constants do not vary across iterations. Then for some constant $\tilde{c}$ which *does not vary across iterations*, for any $L > 0$ and using $\alpha = \frac{1}{2}$, w.p. $1 - \tilde{c}\sum_{j>M}2^{-3j/2}$,

$$\|M_n(\Lambda) - M(\Lambda)\| \leq 2^L n^{-1/3}$$

Figure 3: Mean+95% Monte Carlo CIs of 1000 runs of Mixed Poisson processes, which violate the Poisson process assumption. Synthetic datasets: 20% missingness using AEE wrapped with our method. Black/solid is the true mean function. Red/dotted is initialization with missing data set to Poisson(1) values. Blue/dashed is our EM algorithm initialized at red. (a) $\Lambda^*(u) = \sqrt{u}$ a square root mean function (b) $\Lambda^*(u) = u^2$ a quadratic mean function

Figure 4: Comparison of our EM algorithm to last value carried forward (LVCF), median imputation, and complete case analysis, all under 20% missingness for estimating a step function. We see that last value carried forward performs poorly after a jump and median imputation is a poor method. We show confidence intervals for EM and complete case analysis: the latter has very high variance.

## C   Synthetic Analysis

### C.1   Square Root Synthetic Experiment

We generate synthetic panel count data from mixed inhomogeneous Poisson processes with conditional mean functions $\Lambda^*(u)|X = Xu^{1/2}$, $\Lambda^*(u)|X = Xu^2$, where $X \sim \text{uniform}(0,2)$. The mean functions are then $\Lambda^*(u) = u^{1/2}$ and $\Lambda^*(u) = u^2$, respectively. The counting process conditional on $X$ is Poisson, but the marginal counting process is not. We use 100 trajectories, each with 30 observations and for each observation set it to missing with probability 0.2. We initialize the mean function $\Lambda^{(0)}$ by replacing the missing data with Poisson(1) random variables and fitting a model. We generate 1000 Monte Carlo runs and create Monte Carlo marginal confidence intervals from those runs.

Fig 3 compares the true mean function against AEE wrapped with our method vs AEE directly on the corrupted data. Taking the corrupted data as given learns highly biased results, while wrapping AEE with our algorithm learns close to the true mean function for both experiments.

### C.2   A Step Function Where Two Baselines Perform Poorly

Here we show that three simple baselines perform poorly in estimating a scaled indicator step function. We use last value carried forward, median imputation, and complete case analysis (where we delete each participant with missing data), all under MCAR. Figure 4 shows the results. We see that complete case has very high variance, median is very biased, and LVCF performs poorly after a discontinuity.

# D  Further Analysis of Bladder Tumor Dataset

## D.1  Comment on Mean and Confidence Intervals

In order to form the mean and marginal confidence intervals, we can note that the observation times are all discrete valued at times (months) 1 to 50. Thus we can simply take mean and marginal confidence intervals at those points. To do so, for each bootstrap replicate, we add a small amount of noise $\pm 1e-6$ to the observed time points for identifiability purposes in order to train, and then take mean and marginal confidence intervals at the described points. While last value carried forward is often a surprisingly competitive baseline, it performs poorly after discontinuities, as we see here. Median imputation tends to be highly biased, and complete case analysis tends to have high variance.

## D.2  Varying Missingness Probabilities

In this experiment we vary the missingness probability with $\epsilon = 0.1, 0.2, 0.3, 0.4$. We do so for each of the five following methods: the non-parametric step function maximum pseudo-likelihood (NPMPLE) of [31], the smoothed maximum pseudo-likelihood (MPLs) and and maximum likelihood (MLs) estimators of [19], and the step function solution to the augmented estimating equation (AEE) and the informative censoring (AEEX) version methods of [29]. We show that bias is low but increasing as a function of the missingness probability for the five methods. Figure 5 shows results. The bias is very low in all cases. Further, in all cases it is much lower than the initialization with corrupted data shown in Figure 2a.

## D.3  Varying Initialization

In this experiment we vary the initialization. We note that the mean count across all observations is 0.44. We investigate Poisson(1) to Poisson(4) initializations under both $\epsilon = 0.2$ and $\epsilon = 0.4$. We use AEE in all cases. Figure 6 shows results. We see that further initializations increase bias, but only slightly for $\epsilon = 0.2$ and more so for $\epsilon = 0.4$. This is not surprising as our theory requires good initialization and sufficiently low missingness probability.

## D.4  Heterogeneity in Missingness Probabilities

Even if the MCAR assumption does hold, the missingness probability may vary between subjects. In this experiment, we let $\epsilon | X = \epsilon_{\text{mean}} X$ where $X \sim \text{uniform}(0, 2)$ and $\epsilon_{\text{mean}} = 0.2, 0.4$. This can capture between subject heterogeneity in missingness. Within each subject, we compare initialization of Poisson(1) to Poisson(4). Figure 7 shows results using the AEE method. They look very similar to results without heterogeneity in the missingness probabilities between subjects.

## D.5  Missing at Random

Here we investigate the missing at random (MAR) setting. We compare full data, last value carried forward, median imputation, complete case analysis, EM under MCAR, and EM under MAR. The baselines are under an MCAR assumption. We mimic MAR by artificially masking counts with probability $\epsilon$ conditional on the previous count being zero ($\epsilon = 0.3, 0.4$) or the previous count being non-zero ($\epsilon = 0.2$). Figure 2a) has results, which are generally good and only have a slightly higher bias than the MCAR setting.

## D.6  Missing Not at Random

Here we investigate the missing not at random (MNAR) setting. This is similar to the previous but now we mask counts base on the *current* count being zero or non-zero. Figure 2b) has results. Again, the baselines are under an MCAR assumption. An increase in the zero missingness probability of 1% leads to an approximately 0.28% increase in the final estimate of expected tumors over MCAR: this is still useful for moderate differences in conditional missingness probabilities. For large deviations in the difference in missingness probability, further work is needed.

Figure 5: results for mean and 95% CIs for 1000 bootstrap replicates for bladder tumor dataset, with the missingness probability $\epsilon$ set to $\epsilon = 0.1, 0.2, 0.3, 0.4$. We see that bias is very low in all settings, although it increases with the missingness probability. The methods wrapped with our method are (a) maximum pseudo-likelihood (MPL) of [31] (b) augmented estimating equations (AEE) of [29] (c) augmented estimating equations with informative censoring (AEEX) of [29] (d) maximum pseudo-likelihood splines (MPLs) of [19] note that for this we need to reduce the time axis slightly as spline models cannot interpolate far past the region where they have values in learning (e) maximum likelihood splines (MLs) of [19], same issue.

Figure 6: Here we vary the initialization by initializing increasingly far from the mean observed under complete data. The true sample mean of all intervals is $0.44$. We initialize to Poisson random variables with means $1$ to $4$. We see that initializing further from the truth increases the bias, and that it is worse for the higher missingness probability of $0.4$. Our theory requires good initialization and the missingness probability to be sufficiently low, so this is not surprising.

Figure 7: Here we let the missingness probability vary per participant while still being MCAR by multiplying the mean missingness by some value drawn from uniform$[0, 2]$. We again initialize to Poisson$(1)$ to Poisson$(4)$ for a) mean missingness $0.2$ and b) mean missingness $0.4$.

Figure 8: a) Missing at random (MAR). The time-varying missingness probability is $\epsilon_j = 0.3, 0.4$ if the previous $\Delta N_{j-1} = 0$ (approximately 84% of observations) and $\epsilon_j = 0.2$ otherwise. b) Missing not at random (MNAR): same but instead of using the lagged term $\Delta N_{j-1} = 0$ we use the current $\Delta N_j = 0$. We show confidence intervals for EM and complete case only in order to illustrate the high variance of complete case analysis. Note that for all of these the LVCF, median, and complete case analysis are under an MCAR missingness mechanism.

# E   Further Exploratory Analysis of the Smoking Cessation Study and Dataset

In this section we plot several histograms and boxplots to get insight into behavior variability in the study. We plot cigarettes since the last assessment and number of days between EMAs, where for both we aggregate both between and within subjects. We also plot the percentage of long intervals between subjects. That is, we take the percentage for each participant and then take the boxplots and histograms with the summary statistic for each participant as a data point.

Figure 9 shows the plots for number of cigarettes since the last assessment. We see that most of the time they don't smoke any cigarettes. The histogram looks like a geometric distribution for the number of cigarettes. The mean number of cigarettes is 1.76, the median is 0, and the 25 and 75 percentiles are 0 and 2, respectively. Thus, in at least half of the intervals they don't smoke, and in 75% percent of them they smoke at most two cigarettes, but in some cases they smoke some huge number: in one case someone smoked over $40$ cigarettes. That said, the number of cigarettes also will depend on the length of time elapsed since the last assessment, and the intensity function is time-varying: particularly, we noted in Figure 2b) that they tend to smoke more frequently in the pre-quit period.

Figure 10 plots the days between EMAs. Here we see that most observations are under one day, but there are a substantial number of outliers. In a number of cases the time between EMAs is over two days, and in several cases it is over six days. The mean time between observations is 0.34 days or approximately eight hours, the median is 0.17 or approximately four hours, and the 25 and 75 percentile are approximately two and 11 hours, respectively. Summarizing, most inter-EMA durations are under one day, but a few are very long.

Finally, also relevant is whether there is heterogeneity in the proportion of long intervals between subjects. Figure 11 investigates this. We see that participants vary between having no unreliable intervals and in one case having all unreliable intervals. The mean is 11%, the median is 6%, and the

Figure 9: a) boxplot b) histogram for number of cigarettes since the last assessment. We see that the median number of cigarettes is 0, and that there are many outliers.

Figure 10: a) boxplot b) histogram for number of days between EMAs. Recall that we treat intervals over one day as missing/unreliable. From the boxplot we see that the mean and 75% percentile are under one day, but that there are a substantial number of observations over one day. The histogram suggests a similar finding.

25 and 75 percentile are 1.8% and 11.4%, respectively. Thus there is substantial heterogeneity. While our theory does not currently account for this and we leave it to future work, the experiments from the previous section where each subject had a different missingness probability (Figure 7) suggests that our model can handle this issue well.

Figure 11: a) boxplot b) histogram for percent of long intervals between subjects. We see that most participants have a small proportion of long intervals (the median is 6.1%), while a few have a much larger proportion of long intervals.