[Reviews · NeurIPS 2020]

Review 1

Summary and Contributions: The paper deals with the problem of missingness in panel count data. Specifically, an algorithm for pointwise estimation of the mean function of a Poisson point process in the case of censored counts is presented. The idea is to exploit the expectation-maximization scheme to overcome the problem of missingness: at every E step of the functional EM algorithm, the missing counts are substituted by the mean counts according to the mean function approximation from the previous M step. Data are assumed to be missing-completely-at-random. After the presentation of the algorithm, the work is devoted to provide necessary hypotheses to ensure convergence. In a variational framework (Gateaux derivatives), the contractiveness of the procedure is proved, with specific assumptions on variation of the target function and distance of the initialization. A rate of convergence in probability is given for the sample algorithm. Moreover, the paper includes examples of application to synthetic data, real data with synthetic missingness, and real censored data. In the first analysis, a deviation from the Poisson distribution for the generating counting process is explored. In the second a (slight) deviation from the complete randomness hypothesis is studied. The real data consist in ecological momentary assessment about the number of smoked cigarettes by smoke-quitters since the last request. In this context, long interval reports are considered unreliable by behavioral scientists. Indeed, an estimation of the mean function treating long intervals counts as missing (using the presented algorithm), highlights an (expected) under-reporting. I have read the authors' feedback, and appreciated their effort to elaborate further on the aspects highlighted in the reviews. I confirm my original mildly positive evaluation.

Strengths: The work extends the relatively limited state-of-the-art methodology to process panel count data in presence of missing counts. It provides theoretical guarantees for EM procedures when missing or unreliable records affect panel counts, which may be the case in many real datasets. The motivation for treating data coming from self-report via ecological momentary assessment is interesting and up to date. The work is a straightforward and needed contribution that facilitates the investigation of panel count data.

Weaknesses: The relevant contribution of the work is limited to finite-samples guarantees of the point estimation of the mean function. No uncertainty quantification can be conducted with the proposed method, which is a major drawback. The results are based on many simplifying assumptions. However, the main limits are recognized in the paper. The less convincing argument regards the performance of the procedure under deviations from the missing-completely-at-random assumption. Such deviations are investigated through a single example where the missing generation process slightly deviates from the missing-completely-at-random assumption, leaving many doubts on the performance of the procedure in more realistic scenarios. Moreover, it would have been useful if the authors had provided more insights, as well as a verification procedure, on the conditions that links c, the uniform lower bound on the increments of the true mean function, and r, the radius of the ball inside which the function initialization should lie.

Correctness: Statements and proofs are, overall, correctly carried out. A minor remark for first line of Proof of Lemma 1, in the supplementary material: after the first inequality, the summing index j should not be there anymore, is there a max missing?

Clarity: The paper is overall clearly written. Some minor remarks are the following. - The symbol $\tau$ is used both for endpoint of the time interval and for the missingness flag vector. - The notation $\Delta N^{(\tau)}$ for the elementwise product of $\Delta N$ and $\tau$, resulting in a vector of counts and zeros whether the count is missing, (or the complementary $\Delta N^{(s)}$) is never used again. - Assumption 7 in Section 5.1 is the only one missing an explanation (except in the supplementary material, where it is used in proofs) that may address its acceptability in applications.

Relation to Prior Work: The authors are giving full credit to what is already treated both in the panel count data literature and in the EM convergence one, also in matter of proving techniques.

Reproducibility: Yes

Additional Feedback: The work appears properly carried out and provides a few important results. However, the main idea of using the EM procedure on missing data is not original, the convergence results exploit many assumptions, and the empirical evaluations used to check deviations from the assumptions are limited. Interesting improvements would be: the introduction of some uncertainty quantification technique (to perform tests and compute confidence bounds), a deeper investigation of the performance under other and more realistic missingness mechanisms, and the introduction of some procedure to check whether the number of missing points in a certain dataset is too high to apply the proposed technique.


Review 2

Summary and Contributions: This submission introduces an EM algorithm to estimate the mean parameter of a Poisson process in the presence of missing values. The main motivation is incomplete panel count data, where users self-report a health-related behavior (such as smoking a cigarette after quitting) using an app. The submission provides asymptotic and finite sample convergence properties for the estimator and illustrate its behavior on two datasets.

Strengths: The submission is well written, very clearly presented and easy to follow. It provides a lot of intuition about the content of the paper, making it possible to grasp a first general idea without diving into the notation or technical results, and making it easier to understand these technical results in a second step. Dealing with incomplete panel count data seems like an important and unaddressed topic, which this submission is the first to solve.

Weaknesses: As mentioned by the authors, MCAR is a strong assumption. It is mentioned in the discussion and a theoretical assessment or alternative estimators that are more robust to MCAR may be out of scope, but it seems important to at least assess its impact on the estimation empirically. This seems possible, at least on the bladder data, or even on a separate simulation, eg by making the missing time sampling depend on the count. In the experiments, the estimator is compared to a naive baseline where missing values are set to zero, which strongly biases the estimation of the mean. Other baselines such as replacing missing values by the previous report or the median of all reports could be less biased, and it would be useful to include them in the comparison.

Correctness: The proposed method and analysis seem correct.

Clarity: The submission is very clear and well presented.

Relation to Prior Work: The relation to previous work is carefully discussed. The theoretical properties that are proved for the proposed non-parametric EM follow the line of recent parametric results. This extension seems non-trivial.

Reproducibility: Yes

Additional Feedback: I thank the authors for taking the time to do these additional experiments, they answer the minor points that I raised in my review.


Review 3

Summary and Contributions: The paper proposes an Expectation Maximization (EM) algorithm for panel count data that is censored through a Missing Completely At Random mechanism (MCAR).

Strengths: - The setup and problem description is well written. I enjoyed the description of the problem, particularly the description provided of Ecological Momentary Assessment and relating it to the panel count and missing data problem. - The assumptions are clearly stated which makes it easy to determine which parts of the methodology need to be carefully examined and could be improved in future works or conveyed to collaborators before they decide to use the method. - Principled methods for handling missing data are important.

Weaknesses: - The MCAR assumption is difficult to justify in practice. - Related to the MCAR assumption: Assumption 8 naturally states that the probability of missingness \epsilon is > 0 and from the definition earlier in the paper (it may be better to restate this bound in the assumptions section) that it is also strictly less than 1. - The bound 0 < \epsilon < 1 and MCAR together imply that any parameter of interest in the full law (that can be expressed as a function of the full law) is identified. This is good, however, could the authors clarify some of the following points regarding their method in the context of MCAR missingness. (i) Recovery of functions of the full law under MCAR missingness is very simple. By definition, MCAR implies that one can simply ignore any rows of data containing missingness and restricting the analysis to so called "complete cases" will still result in unbiased estimates of the parameter of interest. In light of this, and the bounds on \epsilon implying that there will always be complete cases in the data as n -> \infty (if this were not true, the parameters of interest would not be identifiable) what is the advantage of the proposed EM algorithm over simply doing complete case analysis and using some of the older tools cited in the paper that can be run on complete data. I apologize if I missed this, but it doesn't seem like there's a baseline comparison to such a complete case analysis or to the alternative of directly maximizing the observed data likelihood by integrating according to patterns of missingness. The latter yields the most (statistically) efficient estimates of the parameter of interest if the parametric form is correctly specified. I think both of these techniques are important baseline comparisons to discuss in theory as well as empirically. (ii) Related to the above point, it may be that in any finite sample there are no complete cases. Is it the case then that the EM proposal is meant to do the "best you can with finite samples"? The sample theory in section 5.4 seems to imply that n must still be "large enough". In which case, standard asymptotics from MCAR complete case analysis and direct maximization of the observed data likelihood should also apply. That is, both of these yield unbiased estimates with the latter being the most efficient way of obtaining the estimates. - I'm also unsure why the work needs to be restricted to MCAR mechanisms, and actually think it should be extensible to more complicated mechanisms without (much) change. That is, I don't think there are any portions of the proofs in the Appendix that would fail had it been a more complicated mechanism such that the full law is still recoverable from the observed data (however, the appendix is fairly lengthy and I may have missed some things). EM has been applied to many MAR problems and actually works for several MNAR mechanisms as long as the assumptions on the missingness process are clearly stated and these assumptions yield identified full laws. Examples of such works are from the graphical modeling literature of missing data processes listed below. [1] Full Law Identification in Graphical Models of Missing Data: Completeness Results (ICML 2020) https://arxiv.org/pdf/2004.04872.pdf [2] Identification In Missing Data Models Represented By Directed Acyclic Graphs (UAI 2019) https://arxiv.org/pdf/1907.00241.pdf [3] Graphical Models for Inference with Missing Data (NeurIPS 2013) https://papers.nips.cc/paper/4899-graphical-models-for-inference-with-missing-data.pdf The relevant point is that in these works, the missingness mechanism (MCAR, MAR, or MNAR) is modeled explicitly and if the proposed missingness mechanism implies that the full law is identified (see [1] for details), this means that maximization of the observed data likelihood (either through EM or by direct maximization of the observed data likelihood) will recover the true parameter of interest. So one nice way of extending the present work may be to posit a set of assumptions on the missingness mechanism using a graph (the underlying distribution need not be graphical, just the missingness process) as in the works listed above or by positing a set of algebraic assumptions that yield identification but these are often harder to convey to clinical collaborators. This is also probably why in D.5 of the Appendix the authors' experiments with MAR mechanisms still yield good results for recovery of the parameters of interest because when the data are MAR, the full law is always identifiable as a function of the observed law. For MNAR mechanisms however, one needs to be more careful as discussed in the works listed above and designing an EM procedure does get a bit more complicated because the identifying functionals are also more complicated. - I think the authors should be more ambitious and extend their methods beyond the MCAR setting because of the reasons stated above (complete case analysis being sufficient for MCAR and the present work possibly being extensible to identifiable MAR/MNAR without much change). This may require significant rewriting though and that plays a major part in my final score because conferences don't have a major revisions option. I think the paper will be quite strong after revisions to incorporate more complicated missingness mechanisms.

Correctness: - I did my best to check the claims and they seem fine to me.

Clarity: - As mentioned above I thought the introduction was quite well written. - Some other parts of the paper could do with reorganization though. There are still definitions being introduced towards the end of the paper. Of course this is a difficult balancing act given the 8 page format but I think the authors could consider restructuring so that technical definitions and intro are done by page 4/5 and there's more space given to intuitions for the proofs as the entirety of these proofs are currently restricted to a fairly lengthy supplement.

Relation to Prior Work: - Some of the work regarding results on the EM algorithm outside of the missingness component were a little confusing in terms of where the authors work began and where previous works ended. It may be good to state more explicitly what has already been shown and which specific portions are new in this paper. - Outside of the works I listed above, there are some other works in the missing data literature that may be useful to mention: [4] Block-Conditional Missing at Random Models for Missing Data (Statistical Science 2010) https://arxiv.org/pdf/1104.2400.pdf [5] Itemwise conditionally independent nonresponse modeling for incomplete multivariate data (Biometrika 2017) https://arxiv.org/pdf/1609.00656.pdf [6] Semiparametric Inference for Non-monotone Missing-Not-at-Random Data: the No Self-Censoring Model https://arxiv.org/pdf/1909.01848.pdf

Reproducibility: Yes

Additional Feedback: - The citations need some fixing. In many places EM is not capitalized appropriately. - While I appreciate the authors' honesty in their usage of MathStackExchange, it may be better to formalize the citation by referring to the original sources found in the StackExchange post i.e. references to the mean value theorem and Darboux's theorem and explicitly invoking these theorems instead of saying "which we can do by [9]" in the proof of Claim 2 where [9] simply refers to the MathStackExchange post. I may even suggest adding the reference to both the theorems as well as the MathStackExchange post. Additional references never hurt! ***************** Feedback after authors response and reviewer discussion ***************** - I have updated my score to a 5 to reflect that I appreciate the theoretical contributions to nonparametric finite sample EM theory. - However, I agree with other reviewers that the contribution in terms of missing data remains minimal if the theory can only be extended to MCAR data at present. - With regards to this, I believe the paper needs a fair amount of revision outside of the scope of this review process in order to redirect emphasis to results that are not currently reflected as being the primary contribution. That is, as it stands, the proposal is presented as a solution to missing data problems. However, it may be better to revise the manuscript to emphasize the nonparametric EM theory as a stand alone component, with applications to missing data as a specific use case. - Other points that I don't think were clarified fully in the author response include maximization of the observed data likelihood. I did not really grasp what the implications of the authors stating that there is no unique maximizer meant. If parameters of a model are identified, then there should be a unique global maximizer. If the parameters are not identified, then EM or any other technique will also face the same issues. That is, I think if there is no information in the likelihood about the parameter of interest, EM and other techniques cannot help learn the parameter. I think the authors should discuss identification in a little more detail.


Review 4

Summary and Contributions: The authors proposed a functional EM algorithm to estimate the mean function for incomplete panel count data. By extending EM algorithm to non-parametric settings, the authors provided finite sample convergence guarantees.

Strengths: Missing value imputation is a key problem in many areas (e.g., healthcare). For this reason, the problem discussed in this paper is very important. The theoretical convergence guarantees are quite interesting, although some key assumptions (e.g., MCAR) are strong and usually do not hold in reality.

Weaknesses: Besides the concerns about the assumptions (discussed in Strengths), my other major comment is related to experiments. 1. It seems that the proposed approach was not compared against any existing imputation method. Without doing so, it is very difficult to see the real value of the work. 2. In Section 6.1, the authors "artificially delete intervals completely at random with probability 0.2.". I am wondering how 0.2 was chosen? Based on my experiences, the probability could be much higher than 0.2 in reality. How does the proposed method work when it is the case? Some parameter analysis (the robustness of the work with respect to the probability) would be nice. 3. I find the results in Section 6.2 a bit weak. Do the results echo findings reported in the literature? Such comparisons could be useful.

Correctness: The theoretical analysis seems to be correct, although I did not investigate this thoroughly.

Clarity: The paper is quite well written.

Relation to Prior Work: It seems that important prior work have been discussed.

Reproducibility: Yes

Additional Feedback: While I appreciate the authors' feedback (adding experiments to address MNAR and to compare with rival methods), I feel that a theoretical analysis of how the proposed method performs under MNAR is lacking. Moreover, as I mentioned in the earlier review, a parameter analysis is necessary to evaluate the robustness of the method. For the above reasons, I would keep my score (marginally below the threshold of acceptance) unchanged.

[Author Response · NeurIPS 2020]

**Summary:** We thank the reviewers for constructive comments, including noting that the work is non-trivial and is a needed contribution. We (re-)emphasize that the key contribution of the present work is the new *finite sample* consistency guarantee for a *non-parametric* EM algorithm; the key technical advance is the novel use of Gateaux derivatives to replace inner products in classical parametric EM theory. **Experiments:** We conducted experiments to address comments about the more challenging MNAR setting and comparison against additional baselines: we conclude that the utility is not limited to MCAR. We mimic MNAR in the bladder tumor dataset by artificially masking counts with probability $\epsilon$ conditional on zero counts ($\epsilon = 0.25, 0.3, 0.35$) or non-zero counts ($\epsilon = 0.2$). Figure 1a) has results. An increase in the zero missingness probability of 1% leads to an approximately 0.28% increase in the final estimate of expected tumors over MCAR: this is still useful for moderate differences in conditional missingness probabilities. For baselines, we include last value carried forward (LVCF), median, and complete case analysis (CCA). At study end, MCAR overestimates the expected number of tumors relative to using the full dataset by $0.11$ ; LVCF underestimates by 0.08 and median and CCA both underestimate by 1.36. CCA has high variance. LVCF does well estimating the bladder tumor mean function, but handles discontinuities poorly. Figure 1b) shows results on a step function: LVCF shows higher bias after a discontinuity than EM. CCA again displays high variance. **Theory:** We clarify the non-trivial nature of theoretical extensions under departure from MCAR, for which the present work paves the way (line 32).

**Reviewer 1: Lack of uncertainty quantification:** this is challenging due to lack of asymptotic normality of the distance between the estimator and true mean function. We took the common approach of proving consistency and rate of convergence [18,30], leaving test statistics to future work [2]. **Linking $c$ to $r$ intuitively:** as the lower bound on expected number of cigarettes over any interval increases, we become more robust to initialization. This happens as minimum smoking risk and/or minimum interval sizes grow. Verifying that $r \leq \frac{c}{4}$ holds in practice requires knowing the true mean function. We recommend (line 135) trying multiple initializations and examining likelihood. Existing EM approaches [3, 35] have similar limitations. **Assumption 7:** is satisfied if $N(\tau)$ (e.g., cigarettes over study) is uniformly bounded. **Lack of missingness tolerance check:** we have this. Our theory holds for $\epsilon < \frac{c}{3b+c}$ (line 219). We need scientific intuition about $b$ and $c$ (uniform upper and lower bound on mean function increments) to apply it.

**Reviewer 2** Thank you for your positive comments. We added two requested experiments in Figure 1.

**Reviewer 3: Complete case analysis baseline:** this requires deleting a participant's data starting from their first missing observation. With 5% missingness and four EMAs per day this on average loses all information after day 5 of a 15 day study, which is inefficient. Figure 1 shows the high variance of this method. **Observed data likelihood baseline:** it may not have a unique maximizer. Consider only one participant with three intervals with the middle missing. The observed data likelihood does not have a unique maximizing mean function (middle increment could be any non-negative value). Adding more participants with no observation times aligning with those of the first participant would still lack a unique maximizer. **Should show MAR and/or MNAR theory:** While asymptotic EM MAR results exist, the finite sample case is unstudied. Current finite sample results assume MCAR [3,35]. Our finite sample MCAR results for panel count are useful for smaller mHealth datasets. In our setting, proving local uniform strong concavity is difficult under MAR. Eqn 13 of the appendix relies on a constant $\epsilon$ and linearity of expectation. **MAR and sometimes MNAR full data distribution is identified:** thank you for the helpful references, which we will cite. Their setting is different: they have multiple observations of the same random variable. In ours, when sampling interval sizes from an absolutely continuous distribution, with probability 1 no two interval sizes across participants will be equal. Every count observation in the study may come from a different random variable. Further the references focus primarily on identification, but estimation poses additional challenges. (Malinsky et al. 2020) addresses estimation, but again in the setting of iid samples of the same random vector.

**Reviewer 4**: Baselines added (Fig. 1). **Lack of sensitivity to missingness:** we have this in Figure 4 of the supplement. **Comparison to literature:** work is limited on whether participants underestimate/overestimate their smoking count EMAs over long intervals. However our psychology coauthors who specialize in smoking consider this finding plausible.

Figure 1: a) MNAR: Missingness ($\epsilon = 0.2$ for non-zero counts, varied $\epsilon$ for zero counts) results in minimal bias for our method. Other baselines treat the MCAR case. b) Step mean function: LVCF introduces bias compared to EM (blue).

[Meta-Review · NeurIPS 2020]

The reviewers all agree that this paper represents a contribution to theory and methods for missing data but note a few limitations. The MCAR assumption is a strong one, but the authors address this in their rebuttal and hopefully more directly in the revised paper. Similarly, it appears they've added baseline performance metrics and other comparisons as requested.